# Encoding a magic state with beyond break-even fidelity

Riddhi S. Gupta[1,2], Neereja Sundaresan[1], Thomas Alexander[1], Christopher J. Wood[1], Seth T. Merkel[1], Michael B. Healy[1], Marius Hillenbrand[3], Tomas Jochym-O'Connor[1,2], James R. Wootton[4], Theodore J. Yoder[1], Andrew W. Cross[1], Maika Takita[1] & Benjamin J. Brown[1,5✉]

To run large-scale algorithms on a quantum computer, error-correcting codes must be able to perform a fundamental set of operations, called logic gates, while isolating the encoded information from noise[1–8]. We can complete a universal set of logic gates by producing special resources called magic states[9–11]. It is therefore important to produce high-fidelity magic states to conduct algorithms while introducing a minimal amount of noise to the computation. Here we propose and implement a scheme to prepare a magic state on a superconducting qubit array using error correction. We find that our scheme produces better magic states than those that can be prepared using the individual qubits of the device. This demonstrates a fundamental principle of fault-tolerant quantum computing[12], namely, that we can use error correction to improve the quality of logic gates with noisy qubits. Moreover, we show that the yield of magic states can be increased using adaptive circuits, in which the circuit elements are changed depending on the outcome of mid-circuit measurements. This demonstrates an essential capability needed for many error-correction subroutines. We believe that our prototype will be invaluable in the future as it can reduce the number of physical qubits needed to produce high-fidelity magic states in large-scale quantum-computing architectures.

We distil magic states to complete a universal set of fault-tolerant logic gates that is needed for large-scale quantum computing with low-density parity-check code architectures[13–18]. High-fidelity magic states are produced[9–11] by processing noisy input magic states with fault-tolerant distillation circuits; experimental progress in preparing input magic states using trapped-ion architectures is described in refs. 3,7. It is expected that a considerable number of the qubits of a quantum computer will be occupied performing magic-state distillation schemes and, as such, it is valuable to find ways of reducing its cost. One way to reduce the cost is to improve the fidelity of input states[11,19–26], such that magic states can be distilled with less resource-intensive circuits.

Here we propose and implement an error-suppressed encoding circuit to prepare a state that is input to magic-state distillation using a heavy-hexagonal lattice of superconducting qubits[4,5,27]. Our circuit prepares an input magic state, which we call a CZ state, encoded on a four-qubit error-detecting code. We explain how our encoded magic state can be used in large-scale quantum-computing architectures[11,28] in the section 'Using CZ states in large-scale quantum-computing architectures'. Our circuit is capable of detecting any single error during state preparation, as such, the infidelity of the encoded state is suppressed as $\mathcal{O}(\varepsilon^2)$, where $\varepsilon$ is the probability that a circuit element experiences an error. By contrast, a standard encoding circuit prepares an input state with infidelity $\mathcal{O}(\varepsilon)$. Furthermore, we can improve the yield

of the prepared magic states with the error-suppressed circuit using adaptive circuits that are conditioned in real time on the outcomes of mid-circuit measurements. We propose several tomographical experiments to interrogate the preparation of the magic state, including a complete set of fault-tolerant projective logical Pauli measurements that can also tolerate the occurrence of a single error during readout.

## Magic-state preparation and logical tomography

We prepare the CZ state as follows:

$$|CZ\rangle \equiv \frac{|00\rangle + |01\rangle + |10\rangle}{\sqrt{3}},$$

encoded on a distance-2 error-detecting code, in which the distinct bit strings label orthogonal computational basis states over two qubits. We can achieve the CZ state by, first, preparing the $|++\rangle = \sum_{a,b=0,1} |ab\rangle/2$ state and, then, projecting it onto the $CZ = +1$ eigenspace of the controlled-phase (CZ) operator $CZ = \mathrm{diag}(1, 1, 1, -1)$, that is, $|CZ\rangle \propto \Pi^+ |++\rangle$ with the projector $\Pi^+ = (1 + CZ)/2$. We can perform both these operations with the four-qubit code. Specifically, it has a fault-tolerant preparation of the $|++\rangle$ state and, as we will show, we can make a fault-tolerant measurement of the logical CZ operator to prepare an encoded CZ state.

[1]IBM Quantum, T. J. Watson Research Center, Yorktown Heights, NY, USA. [2]IBM Quantum, Almaden Research Center, San Jose, CA, USA. [3]IBM Deutschland Research & Development, Böblingen, Germany. [4]IBM Quantum, IBM Research Zurich, Rüschlikon, Switzerland. [5]IBM Denmark, Brøndby, Denmark. ✉e-mail: benjamin.brown@ibm.com

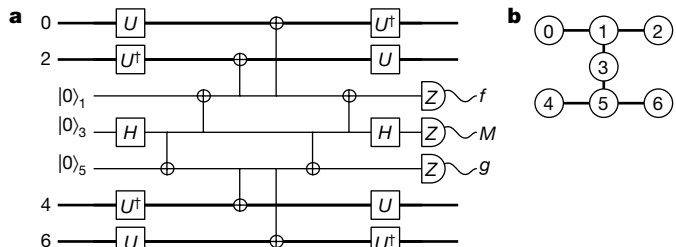

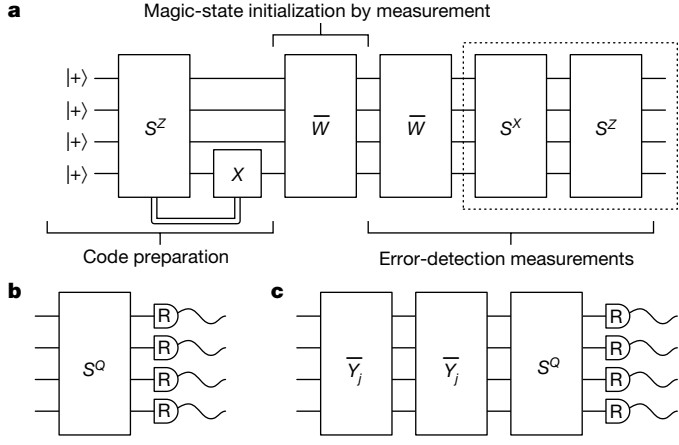

**Fig. 1 | A fault-tolerant circuit to make parity measurements. a**, A circuit that measures $S^X$, $S^Z$ and $\overline{W}$ using flag qubits on the heavy-hexagonal lattice architecture. **b**, The four-qubit code is encoded on qubits with even indices and the other qubits are used to make the fault-tolerant parity measurement. The circuit measures $S^X$ by setting $U = 1$ and $S^Z$ by setting $U = H$, where $H$ is the Hadamard gate. The circuit measures $\overline{W}$ if we set $U = T$. The measurement outcome $M$ gives the reading of the parity measurement. Essential to the fault-tolerant procedure are flag fault-tolerant readout circuits[4,5,27,51] that identify errors that occur during the parity measurement. Outcomes $f$ and $g$ are flag qubit readings that indicate that the circuit may have introduced a logical error to the data qubits.

Encoded states of the four-qubit code lie in the common +1 eigenvalue eigenspace of its stabilizer operators $S^X = X \otimes X \otimes X \otimes X$, $S^Z = Z \otimes Z \otimes Z \otimes Z$ and $S^Y = S^Z S^X$, where $X$ and $Z$ are the standard Pauli matrices. The four-qubit code encodes two logical qubits that are readily prepared in a logical $|\mp \mp\rangle$ state by initializing four data qubits in the superposition state, $|+\rangle \propto |0\rangle + |1\rangle$, and measuring $S^Z$. We note that we use bars to indicate we are describing states and operations in the logical subspace. We prepare the state with $S^Z = +1$ using either postselection or, alternatively, an adaptive Pauli-X rotation on a single-qubit given a random −1 outcome from the $S^Z$ measurement.

The four-qubit code has a transversal implementation of the CZ gate on its encoded subspace, $\overline{CZ} \simeq \sqrt{Z} \otimes \sqrt{Z}^\dagger \otimes \sqrt{Z} \otimes \sqrt{Z}$, where $\sqrt{Z} = \text{diag}(1, i)$. We can measure this operator as follows. We note that conjugating $S^X$ with the unitary rotation $\tilde{T} = T \otimes T^\dagger \otimes T^\dagger \otimes T$, where $T = \text{diag}(1, \sqrt{i})$, gives the Hermitian operator:

$$\overline{W} \equiv \tilde{T} S^X \tilde{T}^\dagger \propto \overline{CZ} S^X. \tag{1}$$

Given that we prepare the code with $S^X = +1$, measuring $\overline{W}$ effectively gives a reading of $\overline{CZ}$.

It is essential to our scheme that we reach the $S^Z = +1$ eigenspace. This is because of the non-trivial commutation relations of $\overline{W}$ with the stabilizer operators of the code[29,30]; $[S^X, \overline{W}] = (1 - S^Z) S^X \overline{W}$. This commutator shows that $\overline{W}$ only commutes with $S^X$ in the $S^Z = +1$ subspace. If $S^Z = -1$, we can check that $\overline{W}$ and $S^X$ anti-commute, and are therefore incompatible observables in this subspace.

We can perform all of the aforementioned measurements, $S^X$, $\overline{W}$ and $S^Z$, on the heavy-hexagon lattice geometry[27]. Figure 1 shows one such setup. The circuit is fault-tolerant in the sense that a Pauli error introduced by a circuit element, on the support of the circuit element, is always detected by a flag qubit or a stabilizer measurement. The verification of this is detailed in the section 'Analysis in terms of single-gate errors'.

We, therefore, present a sequence of measurements that prepare the input magic state and, in tandem, identify a single error that may have occurred during the preparation procedure. Figure 2 shows the sequence and describes its function. As we can detect a single error, we expect the infidelity of the output state to be $\mathcal{O}(\varepsilon^2)$. We compare our error-suppressed magic-state preparation scheme to a standard scheme for encoding a two-qubit magic state, as well as a circuit that prepares the magic state on two physical qubits. Both of these schemes are described in the section 'Standard magic-state preparation circuits'.

We verify our state-preparation schemes by performing two variants of quantum-state tomography to reconstruct the logical state.

**Fig. 2 | Fault-tolerant schemes for magic-state preparation and logical tomography. a**, Preparation of a CZ state on a four-qubit code in three steps. In the code-preparation step, the four-qubit code is prepared in the logical $|\mp \mp\rangle$ state by measuring $|+\rangle^{\otimes 4}$ with the $S^Z$ operator. We can use adaptive circuits or post-selection to correct for $S^Z = -1$ outcomes. In the magic-state initialization step, we measure the $\overline{W}$ operator and post-select on the +1 outcome. In the final error-detection step, we identify the errors that may have occurred during preparation. We measure $\overline{W}$ a second time to identify if a measurement error occurred during the magic-state initialization step. We finally measure $S^X$ and $S^Z$ a second time to identify Pauli errors that may have occurred, and to determine if the initial $S^Z$ measurement gave a readout error. **b,c**, We replace the parity measurements in the dashed box of **a** with circuits **b** and **c** to make logical tomographic measurements and, at the same time, infer a complete set of stabilizer data for error detection. For example, if we set $S^Q = S^X$ and measure qubits in the $R = Z$ basis, we infer the value of $S^Z$, as in **a**, and we also obtain readings of the logical $\overline{Z}_1$, $\overline{Z}_2$ and $\overline{Z}_1\overline{Z}_2$. Likewise, we can set $S^Q = S^Z$ with either $R = X$ to infer $S^X$ as well as logical Pauli operators $\overline{X}_1$, $\overline{X}_2$ and $\overline{X}_1\overline{X}_2$, or $R = Y$ to infer $S^Y$ as well as logical Pauli operators $\overline{X_1 Z_2}$, $\overline{Z_1 X_2}$ and $\overline{Y_1 Y_2}$. In **c**, we include a $\overline{Y}_j$ measurement for logical qubit $j = 1, 2$ to measure logical operators of the form $\overline{Y}_j$, $\overline{Y}_j \overline{X}_k$ and $\overline{Y}_j \overline{Z}_k$ with $k \neq j$ and $k = 1, 2$, where we take an appropriate choice of $R$. The $\overline{Y}_j$ operator is measured twice to identify the occurrence of measurement errors. Operators $\overline{Y}_j$ are supported on three of the data qubits and can therefore be read out with an appropriate modification of the circuit shown in Fig. 1.

The first method uses fault-tolerant circuits that directly measure the logical operators; we refer to this tomographical method as 'logical tomography'. For the second method, which we refer to as 'physical tomography', we perform standard state tomography on the full state of the four data qubits of the system and then project the reconstructed state onto the logical subspace. Logical tomography with the four-qubit code is shown in Fig. 2b,c. All of our logical tomography circuits can tolerate a single error at the readout stage, by repeating the measurement of logical operators and by comparing measurement outcomes to earlier readings of stabilizer measurements.

Logical tomography is more efficient than physical tomography because we are directly measuring and reconstructing the encoded logical state, rather than the physical state. In the case of the four-qubit code, this requires only 7 distinct circuits, whereas physical tomography requires 81 different measurement circuits.

## Experimental results

We performed our experiments using the first-generation real-time control system architecture of IBM Quantum deployed on ibm_peekskill; one of the IBM Quantum Falcon Processors (https://quantum.ibm.com/). Device characterization can be found in the section 'Device overview'. The control system architectures give access to dynamic circuit operations, such as real-time adaptive circuit operations that depend on the outcomes of mid-circuit measurements, that is, feedforward (see section 'Real-time feedforward control of qubits').

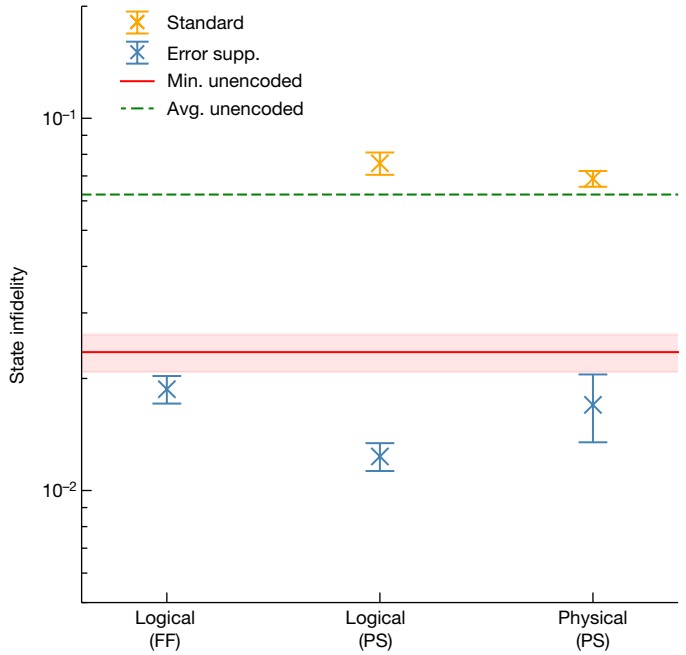

**Fig. 3 | Infidelities measured in magic-state preparation experiments.**
State infidelity for error-suppressed (error supp.) and standard schemes are
shown in blue and orange, respectively. On the *x*-axis, a state is reconstructed
with either logical or physical tomography. The correction for the initial $S^Z$
measurement in Fig. 2a is implemented using either real-time feedforward
(FF) or post-selection (PS). For the physical data points, the state from physical
tomography is projected onto the logical subspace before computing the
infidelity by fitting to ideal projectors. Error bars represent 1σ from
bootstrapping. For all tomographic methods, the error-suppressed scheme
achieves a lower state infidelity compared with the standard scheme. The
unencoded magic state prepared directly on two physical qubits gives an
average (avg.) infidelity across 28 qubit pairs as approximately $6.2 \times 10^{-2}$
(green dashed line) using 18 repetitions over a 24-h period with $10^5$ shots per
circuit. Of these, the best-performing pair yields a minimum (min.) infidelity
of $(2.354 \pm 0.271) \times 10^{-2}$ (red solid line) found over all repetitions for all qubit
pairs. In all cases, the error-suppressed scheme exceeds the fidelity of the best
two-qubit unencoded magic state.

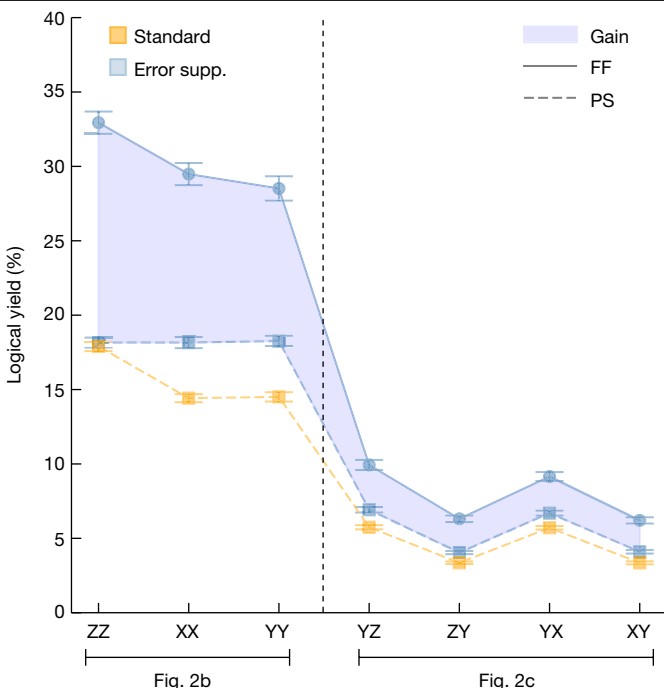

**Fig. 4 | Magic-state yield for feedforward versus post-selection.** Yield is
calculated for logical tomography circuits shown in Fig. 2b,c for the
error-suppressed (error supp.) scheme with feedforward (blue circles) versus
post-selection (blue squares); a standard scheme is shown for reference
(orange squares). All rates use datasets reported in Fig. 3 where error bars
represent 1σ from bootstrapping. The shaded area of the graph shows the
increase in yield for the error-suppressed scheme using feedforward (FF)
compared with the post-selection (PS) scheme or the standard scheme. The
optimal acceptance rate assuming no noise is 75% for the feedforward scheme,
37.5% for the post-selection scheme and 25% for the standard scheme. The
observed acceptance rates are because of the additional detection of errors.
We estimate the yield in the presence of noise in the section 'Estimates for
magic-state yield'. We observe a stark difference in yields between experiments
conducted with the logical tomography circuit shown in Fig. 2b,c, shown to the
left and right of the dashed line, respectively. We can attribute this to the depth
of the logical tomography circuit, in which deeper circuits, such as those shown
in Fig. 2c, are more likely to introduce detectable errors. This is discussed in the
section 'Estimates for magic-state yield'.

Our results are shown in Fig. 3, in which we present state infidelities
for various state-preparation schemes calculated using both logical
tomography and physical tomography. For results provided in the
main text, we model the reconstructed state assuming that the read-
out is conducted with projective measurements. We also present an
alternative analysis in the section 'State tomography with readout error
mitigation using noisy positive-operator-valued measurements', in
which we combine the readout error characterization with tomographic
reconstruction using noisy positive-operator-valued measurements.

To accommodate drift in device parameters over the data collec-
tion period, a complete set of tomography circuits was interleaved
and submitted in batches of about $10^4$ shots until a total of about $10^6$
shots were collected over several days. The resulting counts database
is uniformly sampled with a replacement for 10 bootstrap trials with
a batch size limited to 20% of the total database before post-selection.
The standard deviation, σ, of these bootstrapped trials is plotted as
an error bar in all data figures.

The tomographic fitting was done using positive semi-definite con-
strained weight-least-squares convex optimization using the Qiskit
Experiments tomography module[31]. For logical tomography, the fitting
weights were set proportional to the inverse of the standard errors for
each logical Pauli expectation value estimate. These weights accommo-
date the different logical yield rates for each logical Pauli measurement.

The logical yield for each basis measurement is shown in Fig. 4 and
discussed in more detail below.

We first compare the state-preparation scheme using dynamic
circuits with the same preparation scheme executed with static cir-
cuits and post-selection. This comparison is conducted using logical
tomography. These are the left and middle data points shown in blue
in Fig. 3. We find that the infidelities are commensurate in these two
experiments. Using dynamic circuits with feedforward operations, we
encode a two-qubit error-suppressed input magic state with a logical
infidelity $(1.87 \pm 0.16) \times 10^{-2}$. In the post-selection experiment, we obtain
an infidelity of $(1.23 \pm 0.11) \times 10^{-2}$. The feedforward operations in our
experiment can introduce idling periods, of the order of hundreds of
nanoseconds, during which additional errors can accumulate. To lead-
ing order we attribute the difference in fidelity between these prepara-
tion schemes to errors that occur while the control system is occupied
performing the dynamical feedforward operation. In return for this
loss in fidelity, we find that the use of dynamical circuits significantly
increases the yield of magic states (Fig. 4).

We can analyse the commonly occurring errors in fault-tolerant cir-
cuits using syndrome outcomes to infer the events that are likely to
have caused them[32]. This is done using the method detailed in the

section 'Analysis in terms of single-gate errors' using the results of the error-suppressed scheme without any post-selection. Assuming an uncorrelated error model, we find that the average probability per single-error event is 0.19% with a standard deviation of 0.11%. The single most-likely error event occurs with probability 1.2%. This event corresponds to an error occurring during the $X$ stabilizer measurement that spreads to and is detected by a flag qubit. Similar errors in other stabilizer measurements show the probability increasing from 0.35% for the initial $Z$ measurement to 0.41% and 0.45% for the two $\overline{W}$ measurements. This suggests that, rather than being caused by Pauli errors, these results might be caused by other effects such as an accumulation of leakage on the flag qubits.

We verify the performance of our logical tomography procedure by comparing our results with the infidelity obtained using physical tomography for the magic-state preparation procedure, in which we obtain the $S^Z = +1$ eigenspace with post-selection. The fitter weights in this case are the standard Gaussian weights based on the observed frequencies of each projective measurement outcome of each basis element. In physical tomography, the yield after post-selection is constant in all 81 measurement bases. We find an acceptance rate of $14.9 \pm 0.1\%$ for the error-suppressed scheme using physical tomography, in which the standard deviation represents variation over 81 physical Pauli directions.

To compare the infidelity obtained with physical tomography even-handedly with that obtained using logical tomography, we reconstruct the logical subspace from the density matrix obtained from physical tomography on the data qubits of the code, $\rho_{\text{phys}}$. The logical subspace is obtained by projecting $\rho_{\text{phys}}$ onto the logical subspace[33,34]. We obtain the elements of the density matrix of the logical subspace $\rho$ using the equation

$$\rho_{kl,mn} = \frac{\langle \overline{k}\ \overline{l}|\rho_{\text{phys}}|\overline{m}\ \overline{n}\rangle}{P_L}, \tag{2}$$

where $k, l, m, n = 0, 1$ specify orthogonal vectors in the logical subspace and $P_L = \sum_{k,l} \langle \overline{k}\ \overline{l}|\rho|\overline{k}\ \overline{l}\rangle$ is the probability that the state we prepare is in the logical subspace. Using this method, we obtain the projected logical infidelity for the error-suppressed procedure as $(1.70 \pm 0.35) \times 10^{-2}$ with the probability of finding $\rho_{\text{phys}}$ in the logical subspace $P_L = 0.898 \pm 0.008$. An average post-selection acceptance rate over all physical Pauli directions is found to be $14.9 \pm 0.1\%$. This projected logical infidelity is shown as the rightmost blue data point in Fig. 3 to be compared with the central blue data point. This comparison demonstrates the consistency between logical tomography and physical tomography. For reference, raw state fidelities from physical tomography before logical projection are reported in the section 'State tomography with readout error mitigation using noisy positive-operator-valued measurements'.

We compare our error-suppressed magic-state preparation procedure with a standard static circuit that encodes a physical copy of the magic state into the four-qubit code. We show infidelity data points for the standard scheme in Fig. 3 with orange markers. Our experiments consistently demonstrate that our error-suppressed encoding scheme has an infidelity at least four times smaller than a standard scheme to encode magic states. We show yields using different logical tomography experiments for the standard preparation scheme with orange markers in Fig. 4. In the case of physical tomography, the encoded state on the four data qubits has a post-selection acceptance rate of $20.9 \pm 0.1\%$, and the reconstructed density matrix is found in the code space with probability $P_L = 0.789 \pm 0.004$.

Finally, we compare our error-suppressed preparation procedure with a state-preparation experiment performed using physical qubits. We mark the lowest infidelity obtained over all of the adjacent pairs of physical qubits on the 27 qubit device, $(2.4 \pm 0.3) \times 10^{-2}$, with a red line in Fig. 3. Remarkably, all fidelities for all of our error-suppressed magic-state preparation schemes exceed the fidelity of a simple experiment to prepare the CZ state with physical qubits.

## Discussion

We have presented a scheme that encodes an input magic state with a fidelity higher than we can achieve with any pair of physical qubits on the same device using basic entangling operations. This improvement in fidelity, which is beyond the break-even point set by basic physical qubit operations, can be attributed to quantum error correction that suppresses the noise that accumulates during state preparation.

The yield of magic states benefited from the use of dynamic circuits in which mid-circuit measurements condition gate operations in real time. Remarkably, we find that the operation is sufficiently rapid that its execution came at only a small cost in output state fidelity on the superconducting device. These dynamic circuits are essential to future quantum-computing architectures as they will be needed, for example, to perform magic-state distillation circuits[9–11] and gate teleportation[35,36], as well as many other measurement-based methods[13,17–19,37–48] that have been proposed to complete a universal set of logic gates.

We have shown that experimental progress has reached a point at which we can make prototype gadgets that can affect the resource cost of large-scale quantum computers. In the Methods, we explain how our prototype can be used together with magic-state distillation. It will be interesting to continue to design, develop and test new gadgets with real hardware that will improve the performance of the key subroutines needed for fault-tolerant quantum computing. Further developments in the theory of pieceable fault tolerance[44] might show us ways of producing better magic states with small devices. Error-suppressed magic states could improve the time cost of recent proposals[49,50] for error-corrected circuits that are supplemented by error-mitigation techniques to complete non-Clifford operations. Ultimately, experimental progress that we make to this end in the near term can benefit large-scale quantum-computing architectures.

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

## Methods

### Using CZ states in large-scale quantum-computing architectures

Magic states are distilled to complete a universal set of fault-tolerant logic gates (see Extended Data Fig. 1 for an overview of the details of a generic magic-state distillation protocol). In any such protocol, input magic states with some inherent error are encoded on quantum error-correcting codes. The encoded magic states are then used in distillation circuits to produce better magic states with higher fidelity. We can use magic states with near-perfect fidelity to perform fault-tolerant logic gates.

We choose different magic-state distillation protocols depending on the magic states we prepare. In the next section, we review details on how we can use CZ states in large-scale fault-tolerant quantum computing. Specifically, we can convert CZ states into Toffoli states using Pauli measurements and Clifford operations[28], such that we can adopt well-known magic-state distillation protocols for further rounds of distillation[11]. This conversion technique is probabilistic, as it depends on obtaining the correct outcome from a Pauli measurement. In addition to these results, we show how we can recover a CZ state from the output state, assuming we get the incorrect outcome of the Pauli measurement, thereby conserving the available resource states.

We also provide some examples of how we can inject small codes into larger codes in the section 'Injecting small codes into larger codes', as it will be required to take the encoded state we prepared in the main text and use it in subsequent rounds of magic-state distillation. Specifically, we show how to take distance-2 codes and encode their state on the surface code, the heavy-hex code and the colour code with a higher distance. Notably, the heavy-hex code is readily implemented on the heavy-hex lattice geometry on which we conducted the experiment. For each of these injection schemes, we argue that we can detect any single error that may occur. This is important to maintain the error suppression obtained when preparing the CZ state.

In the case of the colour-code state-injection protocol, we inject the error-detecting code described in the main text directly into the larger code. In the case of the surface code and heavy-hex code, however, we inject a related code, that we call the [[4, 1, 2]] code. To complete these injection protocols, we need to take the CZ state prepared on the error-detecting code and copy its logical state onto two copies of the [[4, 1, 2]] code. We give a fault-tolerant procedure for this transformation in the section 'Encoding the CZ state on two [[4, 1, 2]] codes using the heavy-hex lattice geometry'.

**Magic-state distillation with the CZ state.** Although magic-state distillation for the CZ state has not been well-studied in the literature, it is known that two copies of the state can be probabilistically converted into a Toffoli state using Pauli measurements and Clifford operations[28]. Given there are known methods for distilling Toffoli states[11], let us review how the Toffoli state is produced from the copies of the CZ state. In the following sections, we show how to inject the CZ state into larger quantum error-correcting codes that are capable of performing fault-tolerant Clifford operations[17,18,45] to complete these circuits.

The Toffoli state is defined as follows:

$$|\text{TOF}\rangle \propto \sum_{j,k} |j\rangle|k\rangle|jk\rangle = |000\rangle + |010\rangle + |100\rangle + |111\rangle, \tag{3}$$

where we sum over the bitwise values $j, k = 0, 1$.

Given two copies of the CZ state, $|CZ\rangle_{1,2} |CZ\rangle_{3,4}$, if we project qubits 2 and 3 onto the $Z_2Z_3 = -1$ eigenspace, then we obtain the intermediate state

$$|\xi\rangle = (|0010\rangle + |1010\rangle + |0100\rangle + |0101\rangle)/2. \tag{4}$$

We then obtain $|1\rangle|\text{TOF}\rangle$ with the following unitary circuit

$$|1\rangle |\text{TOF}\rangle = CX_{4,3}CX_{3,1}CX_{2,1}CX_{1,3} |\xi\rangle, \tag{5}$$

where indices $C$ and $T$ of the controlled-not gate $CX_{C,T}$ denote the control and target qubit, respectively.

We obtain the $-1$ outcome by measuring $Z_2Z_3$ of state $|CZ\rangle |CZ\rangle$ with probability 4/9. Beyond the work in ref. 28, we find that we can recover a single copy of the CZ state given the $Z_2Z_3 = +1$ outcome at this step, thereby saving magic resource states. In the event that we obtain this measurement outcome, we produce the state

$$|\chi\rangle = |0000\rangle + |0001\rangle + |1000\rangle + |1001\rangle + |0110\rangle. \tag{6}$$

Applying the unitary operation $CX_{2,3}CX_{2,4} |\chi\rangle$ and obtaining the two-qubit parity measurement outcome $Z_3Z_4 = -1$, we obtain the state $|CZ\rangle |01\rangle$. We obtain this state with probability 3/5, assuming we obtained the $Z_2Z_3 = +1$ outcome previously.

**Injecting small codes into larger codes.** Magic-state distillation takes encoded magic states, and then processes these input states to probabilistically prepare a magic state with better fidelity. As such, it is necessary to encode magic states into quantum error-correcting codes. This process is commonly known as state injection.

Ideally, the injection process will introduce a minimal amount of noise to the logical state that is encoded, as this will reduce the noise of the output magic state. To this end, we look for ways to inject the magic state prepared on the four-qubit error-detecting code in larger quantum-error-correcting codes in such a way that local errors can be detected.

In what follows, we show how to inject the state encoded on the error-detecting code onto the surface code, the heavy-hex code and the colour code, thereby increasing the distance of the code that supports the magic state. Furthermore, we argue that we can detect any single error that may occur during the injection procedure. This enables us to maintain the error suppression we demonstrated experimentally in the main text.

In the main text, we showed how to prepare the CZ state on a four-qubit error-detecting code shown in Extended Data Fig. 2 (left). As we show later, states on this code can be injected directly onto the colour code. Two of the injection schemes, encoding onto the surface code, or the heavy-hex code, assume that the two logical qubits of the CZ state are encoded on two copies of the [[4, 1, 2]] code, shown in Extended Data Fig. 2 (right). In the following section, we show how to encode the magic state prepared on the error-detecting code onto two copies of the [[4, 1, 2]] code, in a fault-tolerant way such that any single error can be detected. For the remainder of this section, we assume the magic state has been prepared over two copies of the [[4, 1, 2]] code.

To distinguish the two small error-detecting codes of interest consistently, throughout the Methods we will refer to the error-detecting code used in the main text as the [[4, 2, 2]] code to contrast this code with the [[4, 1, 2]] code. Specifically, we label the codes by their encoding parameters [[n, k, d]]. Both of these codes have a distance $d = 2$ using $n = 4$ physical qubits. The two codes differ by the number of logical qubits they each encode. The [[4, 2, 2]] code encodes $k = 2$ logical qubits and the [[4, 1, 2]] code encodes $k = 1$ logical qubit.

**The theory of code deformations.** We inject a state into a larger code[11,19–26,41,52] using code deformation[19,46,53]. In what follows, we describe the theory of code deformations using the stabilizer formalism. We remark that more general theories of code deformations can be found elsewhere in the literature[46,53]. The theory we present is sufficient to describe the state-injection operations of interest.

We describe code deformations using the stabilizer formalism[54]. Quantum error-correcting codes can be described with an Abelian

subgroup of Pauli operators called the stabilizer group $\mathcal{S}$. The encoded state lies in the common +1 eigenvalue eigenspace of the elements of the stabilizer group. We call this subspace the code space. Stabilizer codes also have associated logical operators $\mathcal{L}$ that can be generated by a set of mutually anti-commuting pairs $\overline{X}_j, \overline{Z}_j \in \mathcal{L}$ with $1 \le j \le k$. These Pauli operators commute with the stabilizer group but are not themselves stabilizer operators. The distance of the code $d$ is the weight of the least-weight logical operator. We can detect any single error if the code has a distance of at least $d = 2$. We give examples of small stabilizer codes, together with their logical operators in Extended Data Fig. 2. These examples will be relevant for the following discussion on state injection.

We measure the stabilizer operators to identify the errors. As the encoded state is specified by specific eigenstates of a list of commuting Pauli operators, finding a measurement of one or more of these operators in the incorrect eigenspace indicates that an error has occurred. By arguing that we can detect any single error, we must have a distance of at least $d = 2$.

A code deformation is where we perform a measurement that projects a stabilizer code onto another. Specifically, we assume we have prepared an initial code in which, once prepared, we start measuring the stabilizer operators of a second code that we call the final code. This projects the initial code onto the final code. Let us denote these two codes by their stabilizer group $\mathcal{S}_{\text{init}}$ and $\mathcal{S}_{\text{fin}}$, respectively. We assume errors may have occurred on the qubits of the initial state that must be detected by measuring the stabilizers of the final code. As such, this operation has an associated code distance, according to the number of local error events that must occur for an undetectable logical error to affect the encoded space.

We detect the errors by comparing repeated readings of stabilizer measurements. Specifically, once we measure the stabilizers $\mathcal{S}_{\text{fin}}$, we look to compare their outcomes to stabilizers prepared in the initial code $\mathcal{S}_{\text{init}}$. Variations in the values of these stabilizer measurements indicate that an error has occurred. As such we are interested in code-deformation stabilizers

$$\mathcal{S}_{\text{def}} = \mathcal{S}_{\text{init}} \bigcap \mathcal{S}_{\text{fin}}, \tag{7}$$

that is, stabilizers that are prepared in the initial system and checked again after the code deformation is made, when we measure the stabilizer group $\mathcal{S}_{\text{fin}}$.

Logical information that is preserved over the code deformation has coinciding logical operators associated with both $\mathcal{S}_{\text{init}}$ and $\mathcal{S}_{\text{fin}}$. Specifically, the logical operators that are preserved over the code deformation $\mathcal{L}$ are of the form

$$\mathcal{L} = \mathcal{L}_{\text{init}} \bigcap \mathcal{L}_{\text{fin}}, \tag{8}$$

where $\mathcal{L}_{\text{init}}$ and $\mathcal{L}_{\text{fin}}$ are the logical operators for $\mathcal{S}_{\text{init}}$ and $\mathcal{S}_{\text{fin}}$, respectively.

Ideally, we should maximize the number of stabilizers that coincide in the initial and final codes to maximize the number of errors we detect. In practice, physical constraints imposed by hardware may not allow us to maximize the intersection between $\mathcal{S}_{\text{init}}$ and $\mathcal{S}_{\text{fin}}$. Here we concentrate on very simple initialization procedures in which the initial stabilizer code is prepared in a product state, or a product state of Bell pairs, together with the small four-qubit codes that initially maintain the encoded magic state.

**Error correction for state injection.** In what follows, we will show state injection into the surface code, the heavy-hex code and the colour code. We will also argue that all of these state-injection protocols are tolerant to a single error, thereby maintaining the error suppression achieved in the experiment presented in the main text.

We are interested in the general error model, in which a single error occurs on a circuit element in the stabilizer readout circuit as we deform

the initial code onto the final code. However, we argue that for each individual example, we need to study only single-qubit errors that occur immediately before the code deformation takes place.

In addition to the errors that occur on data qubits, we are also interested in errors that occur on the auxiliary measurement qubits we use to perform parity measurements. In essence, these can lead to two types of error: (1) readout errors, in which we obtain the incorrect measurement outcome; and (2) hook errors, in which an error during a stabilizer readout circuit is copied to several other qubits, thereby creating a correlated error. Let us mention how we treat these types of error in the following discussion.

First of all, we neglect to discuss hook errors, as we assume that measures can be taken to mitigate their effects, by either flag qubits or an appropriate choice of stabilizer readout circuit. These measures are well developed for the codes of interest, see, for example, refs. 27,55–57. We completed the experiment presented in the main text using a device that is tailored to realize the heavy-hex code using additional flag qubits to mitigate the effects of hook errors.

We can detect a measurement error using a generic method, namely, the repetition of measurements. By repeating the measurements at least once, we can identify a single measurement error if the outcomes of two repetitions of the same measurement do not agree. As this method is applicable to all of the following injection schemes, we will not discuss this error-detection method case by case. Rather, we argue now that by measuring the stabilizer generators of $\mathcal{S}_{\text{fin}}$ twice we can detect any single error. If the measurements of the two rounds of $\mathcal{S}_{\text{fin}}$ do not agree, we discard the state we have prepared and repeat the state-preparation procedure. Otherwise, assuming the two rounds of measurement for $\mathcal{S}_{\text{fin}}$ do agree, we check the outcomes to determine whether any Pauli errors occurred during the preparation of $\mathcal{S}_{\text{init}}$, or immediately before the $\mathcal{S}_{\text{fin}}$ stabilizer generators are measured. Assuming no error is detected, we continue to conduct standard error correction with the final code.

*Surface code*

Let us start by discussing the example of the surface code[58] (Extended Data Fig. 3). The stabilizers of the code are shown by the faces in Extended Data Fig. 3 (left), in which the light faces mark the support of Pauli-X-type stabilizers and the dark faces mark the support of Pauli-Z-type stabilizers. We also show the support of a Pauli-X logical operator in green and a Pauli-Z logical operator in blue. In the theory for code deformation given above, this is the stabilizer group for $\mathcal{S}_{\text{fin}}$.

In Extended Data Fig. 3 (right), we show $\mathcal{S}_{\text{init}}$. The figure shows the [[4, 1, 2]] code outlined in red in the bottom-left corner of the lattice. The remaining qubits are prepared in a product state, such that the blue qubits are initialized in the $|0\rangle$ state and the green qubits are initialized in the $|+\rangle$ state. These disentangled qubits can be regarded as being in the stabilizer state $Z_v$ or $X_v$. The logical operator of the initial state can be supported entirely on the [[4, 1, 2]] code. However, the initial code shares the logical operators of the final code if we multiply the logical operators of the [[4, 1, 2]] code by the product state stabilizers.

Importantly, all the qubits support at least one stabilizer operator of $\mathcal{S}_{\text{def}}$ such that a single error can be detected. We note that the qubits that are initialized in a product state need to detect only one type of error, because the other type of error acts trivially on the initial state. For example, a Pauli-X error acts trivially on a green qubit and a Pauli-Z error acts trivially on a blue qubit, whereas, respectively, a Pauli-Z or Pauli-X error on the same qubit will be detected by a stabilizer of $\mathcal{S}_{\text{def}}$. Finally, all qubits of the [[4, 1, 2]] code support one of each type of stabilizer of $\mathcal{S}_{\text{def}}$ and as such can also detect both types of Pauli error. By inspection then, we see that we can detect any single-qubit error that occurs at the initialization step.

*Heavy-hex code*

We can also inject the [[4, 1, 2]] code into the heavy-hex code. This is particularly relevant with respect to the experiment presented in

the main text, as the experiment is implemented on hardware that is tailored to realize the heavy-hex code. The heavy-hex code is a subsystem code closely related to the surface code. However, as the code is a subsystem code, stabilizers are not measured directly. Rather, we have a group of check operators, known as the gauge group, that we measure to infer the values of the stabilizer operators. Nevertheless, we find the arguments given above are sufficient to show that a state can be injected while detecting a single error.

To review, the gauge group of the heavy-hex code includes weight-2 Pauli-Z terms on adjacent pairs of qubits that share a row. We show one such term in Extended Data Fig. 4 (left). The code also has Pauli-X-type checks. These are identical to the Pauli-X-type stabilizer operators of the surface code (Extended Data Fig. 3). These checks are used to infer Pauli-X- and Pauli-Z-type stabilizer operators. The Pauli-X-type stabilizer operators are the product of Pauli-X terms on all of the qubits on two adjacent rows (Extended Data Fig. 4, left). The Pauli-Z stabilizer operators are the same as those of the surface code (Extended Data Fig. 3). We also show the support of logical Pauli-X and Pauli-Z stabilizer operators in Extended Data Fig. 4 (left) in green and blue, respectively. Once again, this stabilizer group can be regarded as $\mathcal{S}_{\text{fin}}$ with respect to the simplified code-deformation theory we have presented. Although we infer their values from measuring the gauge checks, the basic theory of state injection holds for our discussion on error correction.

We show $\mathcal{S}_{\text{init}}$ for the heavy-hex code in Extended Data Fig. 4 (right), in which the [[4, 1, 2]] code, highlighted in red, is prepared on qubits in the bottom-left corner of the lattice, and the green qubits are prepared in the $|+\rangle_v$ state and the blue qubits are prepared in the $|0\rangle_v$ state. These qubits have an associated stabilizer $X_v$ or $Z_v$. Once again, similar to the case of surface code, the logical operators that are completely supported on the [[4, 1, 2]] code can be multiplied by elements of the stabilizer group of $\mathcal{S}_{\text{init}}$ such that they are equivalent to those of $\mathcal{S}_{\text{fin}}$ shown in Extended Data Fig. 4 (left). As such, the encoded logical information is preserved over the state-injection procedure, as these logical operators are members of $\mathcal{L}_{\text{def}}$.

As with the case of the surface code, we argue that we can tolerate any single-qubit error during the injection procedure. Every single green qubit supports at least one Pauli-X-type stabilizer and every single blue qubit supports at least one Pauli-Z-type stabilizer of $\mathcal{S}_{\text{def}}$. As such, we can detect a single Pauli-Z error on the green qubits and a single Pauli-X error on the blue qubits that occurs up to the point the code deformation takes place. We are not concerned with Pauli-X errors acting on the green qubits and the Pauli-Z errors acting on the blue qubits as these errors act trivially on the initial state. Finally, all of the qubits of the [[4, 1, 2]] code support both a Pauli-X- and a Pauli-Z-type stabilizer of $\mathcal{S}_{\text{def}}$, and as such, they can all detect both types of error. This accounts for single-qubit errors occurring on all of the qubits of the system during the state-injection process with the heavy-hex code.

### Colour code

Let us finally discuss the colour code[59]. This is a particularly interesting example as the [[4, 2, 2]] code can be injected directly into the colour code. We show the colour-code lattice in Extended Data Fig. 5. For $\mathcal{S}_{\text{fin}}$, each lattice face supports both a Pauli-X- and Pauli-Z-type stabilizer. The code supports two logical operators, where $\overline{X}_A$ is the product of Pauli-X terms supported on all the qubits along the bottom boundary of the lattice and $\overline{Z}_A$ is the product of Pauli-Z terms supported on all the qubits along the left boundary of the lattice. Likewise $\overline{X}_B$ is the product of Pauli-X terms supported on all the qubits along the left boundary of the lattice and $\overline{Z}_B$ is the product of Pauli-Z terms supported on all the qubits along the bottom boundary of the lattice. We highlight the support of the logical operators on the left and bottom boundaries in blue and green, respectively, in Extended Data Fig. 5.

We define the stabilizer group for $\mathcal{S}_{\text{init}}$ in the caption of Extended Data Fig. 5, in which the [[4, 2, 2]] code is placed on a four-qubit face of the lattice, and all of the other qubits are prepared in Bell pairs, with stabilizer operators $X_aX_b$ and $Z_aZ_b$, marked by highlighted edges in the

figure. We colour the edges either blue or green according to the colouring convention for edges used in ref. 59. Nevertheless, all highlighted edges, of both colours, support the same Bell pair.

We can multiply the logical operators of the [[4, 2, 2]] code by elements of $\mathcal{S}_{\text{init}}$ such that we obtain the logical operators of $\mathcal{S}_{\text{fin}}$. As such, the logical qubits encoded on the error-detecting code are preserved over the state-injection process.

We finally argue that we can detect any single-qubit error during the state injection process. Extended Data Fig. 5 shows the support of the stabilizer operators of $\mathcal{S}_{\text{def}}$ with coloured faces. Specifically, there is both a Pauli-X- and Pauli-Z-type stabilizer on each of the coloured faces. By inspection, we see that every qubit supports at least one coloured face and, therefore, supports both a Pauli-X- and a Pauli-Z-type stabilizer. We note also that the error-detecting code also supports both a Pauli-X- and a Pauli-Z-type stabilizer on its respective face. As such, we can detect any single-qubit error over the state-injection process.

**Some remarks on state-injection procedures.** We have presented state-injection protocols for several different codes for the error-suppressed magic state we discussed in the main text. We argued that we can detect a single error that may occur in any of these protocols, such that we maintain the error suppression we have demonstrated in our experiment. The injection protocols we have presented can be improved by combining them with other methods presented in the literature to improve the performance and yield of state injection. For instance, in refs. 20,24, two-step preparation procedures are proposed, in which a magic state is injected onto an intermediate-sized code, where error detection is used to suppress errors, before injecting the intermediate-sized code onto a larger code. This method is compatible with the injection protocols we have presented here. We might also adopt the method presented in ref. 25, in which the authors propose estimating the logical error on the injected state in the decoding step of state injection.

It is worth remarking that these error-detection protocols can be improved by increasing the fraction of error events that can be detected. We might, for example, consider better choices of $\mathcal{S}_{\text{init}}$ that can be prepared before the state-injection procedure begins. In the case of subsystem codes, we might also look for additional error-detection checks that can be made between intermediate gauge measurements we make to infer the values of the stabilizers, and the stabilizers of the initial code, during the preparation procedure.

**Encoding the CZ state on two [[4, 1, 2]] codes using the heavy-hex lattice geometry.** Two of our state-injection protocols described above require that the CZ state is encoded on copies of the [[4, 1, 2]] code. Here we show how to transform the encoded CZ state prepared on the [[4, 2, 2]] code as we have described in the main text onto two copies of the [[4, 1, 2]] code. This transformation is made using measurements. In this sense, it can be understood as a code deformation similar to that discussed in the previous section. We argue that we can detect any one single error over the code deformation process, thereby maintaining the error suppression obtained in the main text. We also show how this process can be mapped onto the heavy-hex lattice geometry. The protocol is outlined in Extended Data Fig. 6, and we show how the outline is mapped onto the heavy-hex geometry in Extended Data Fig. 7.

Before discussing the transformation, we first briefly review the ideas behind state teleportation abstractly. We can view the transformation as a small instance of a lattice surgery operation[38] in which the gates are performed between logical qubits by measuring appropriate logical degrees of freedom. Furthermore, in this particular instance, we can view the operation as a lattice surgery operation between a small colour code and a small surface code[48,60,61], in which we interpret the [[4, 2, 2]] code and the [[4, 1, 2]] code as a small colour code and surface code, respectively. After performing a logical parity measurement

between the two codes, the transformation is completed with a partial condensation operation of the small colour code, as described in ref. 48.

To explain the operation, we consider the evolution of the stabilizers and logical operators of the code at each step of the measurement pattern shown in Extended Data Fig. 6 independently from the implementation of the code. We have three logical qubits indexed $A$, $B$ and $C$, where, initially, $A$ and $B$ are encoded on the $[[4, 2, 2]]$ code and $C$ is encoded on the $[[4, 1, 2]]$ code. In essence, the operation teleports the logical state encoded on qubit $B$ onto qubit $C$, up to a Clifford operation. Logical qubit $A$ is not involved in the operation, so we concentrate on qubits $B$ and $C$.

The teleportation operation proceeds as follows:
1. Prepare $|+\rangle_C$,
2. Measure $X_B Z_C$,
3. Measure $Z_B$,
4. Apply Pauli correction.

The operation functions with $A$ and $B$ prepared in some arbitrary logical state, but to illustrate the operation we assume they are in a product state with $|\psi\rangle_B = a |+\rangle_B + b |-\rangle_B$. We omit qubit $A$ from the discussion, as it is unchanged by the transformation, and we leave it as an exercise to the reader to verify the general case.

Initially, an arbitrary state in the $B$ subsystem along with a logical $|+\rangle$ state on the $C$ subsystem can be described by the following vector state: $(a |+\rangle + b |-\rangle)_B \otimes |+\rangle_C$, in which we have chosen a convenient basis for the vectors on $B$. Upon measuring the joint logical operator $X_B Z_C$ and obtaining measurement outcome $m_2$, the resulting state of the joint system is $a |+\rangle_B |m_2\rangle_C + b |-\rangle_B |1 \oplus m_2\rangle_C$. Finally, upon measuring $Z_B$ and obtaining the measurement outcome $m_3$, the resulting state is $|m_3\rangle_B \otimes (a |m_2\rangle_C + (-1)^{m_3} b |1 \oplus m_2\rangle_C$. An appropriate Pauli correction depending on the measurement outcomes $m_2$ and $m_3$ enables us to recover the state $|0\rangle_B \otimes (a |0\rangle_C + b |1\rangle_C)$. As such, we see the logical information that was originally encoded on the $B$ subsystem in the form of the coefficients $a$ and $b$ now lies entirely on the $C$ subsystem. Finally, we note that, with this operation, the basis of the logical information has been rotated by a Hadamard operation. This can be corrected at a later step. Extended Data Fig. 6 shows how this transformation is conducted between an encoded qubit of the $[[4, 2, 2]]$ code and the logical qubit of the $[[4, 1, 2]]$ code by performing logical measurements.

We now discuss how to implement the described state teleportation on a heavy-hex lattice (Extended Data Fig. 7). We begin by preparing the encoded CZ state as explained in the main text, together with an encoded $[[4, 1, 2]]$ code. The $[[4, 1, 2]]$ code is prepared in the logical state $|+\rangle$. We can prepare this state using qubits outlined in the orange box shown in Extended Data Fig. 7 (top), in which the four qubits, 4, 6, 15 and 17 are the data qubits of the code and qubits 5, 10 and 16 are used to perform weight-4 parity checks with qubits 5 and 16 used as flag qubits. The $[[4, 1, 2]]$ code is prepared in a fault-tolerant manner by initializing the data qubits in the $|+\rangle$ state and then measuring each of the Pauli-Z-type stabilizer operators $Z_4 Z_6$ and $Z_{15} Z_{17}$. These measurements can be facilitated with the ancillary qubits 5 and 16, respectively. Each of these operators is measured twice such that we can detect a single measurement error during preparation (see also ref. 4).

We transfer a single logical qubit of the $[[4, 2, 2]]$ code onto the $[[4, 1, 2]]$ code using logical measurements. In step 3, we perform a weight-4 measurement that measures the parity of two logical qubits over the two codes. To do this using the heavy-hexagonal lattice geometry, we first transport the codes. This can be performed in two rounds of swap gates or teleportation operations, as shown by the arrows in Extended Data Fig. 7 (top), in which the blue arrows are performed first, in parallel, and the green arrows are performed in parallel afterwards. It should be noted that these rounds of parallel swap gates are fault-tolerant because all individual swap operations involve a single data qubit as well as an ancillary qubit. Thus, any potential two-qubit

gate error is effectively a single-qubit error on the code that would be detected. After the swap operation, we facilitate the logical parity measurement with qubits 5, 10 and 16, shown in the green box in Extended Data Fig. 7. The logical measurement is performed twice to identify a measurement error that may occur in this step. The outcomes of both of these measurements should agree. An odd parity in measurement outcomes indicates that a measurement error has occurred.

Finally, we measure the logical operator $\overline{Z}_B$ to complete the teleportation operation. We measure this operator on both of its two-qubit supports. Specifically, these are $\overline{Z}_B = Z_2 Z_4$ and $S^Z \overline{Z}_B = Z_{13} Z_{15}$, in which $S^Z = Z_2 Z_4 Z_{13} Z_{15}$ is the weight-4 Pauli-Z stabilizer of the $[[4, 2, 2]]$ code. Measuring both of these weight-2 logical operators enables us to detect a single error, as the product of their outcomes should agree with the value of the Pauli-Z stabilizer $S^Z$. This final measurement completes the teleportation operation and, moreover, projects the error-detecting code onto a second copy of the $[[4, 1, 2]]$ code. Finally, we remark that projecting $\overline{Z}_B$ into a known eigenstate enables us to regard this logical operator as a weight-2 stabilizer. As such, we can now regard the $[[4, 2, 2]]$ code that we prepared initially as a $[[4, 1, 2]]$ code. We, therefore, have the state $\mathbb{1} \otimes H |CZ\rangle$ encoded on the logical space of two $[[4, 1, 2]]$ codes shown in the purple and orange boxes shown in Extended Data Fig. 7 (bottom).

### Analysis in terms of single-gate errors

All circuits considered, both for magic-state preparation and logical tomography, use redundancy to detect errors. For the mid-circuit syndrome measurements, performed with the circuit shown in Fig. 1, this redundancy comes, in part, by using flag qubits. These yield an outcome of 0 unless an error has occurred. These outcomes are, therefore, error-sensitive events, allowing errors to be detected.

Additional error-sensitive events come from the results of the syndrome measurements themselves. For the circuit shown in Fig. 2b, these events are as follows:
1. The results of the two $\overline{W}$ measurements should agree.
2. $S^X$ should yield 0, because the system is prepared in a +1 eigenstate of this operator.
3. Although the first $S^Z$ will yield a random outcome, the following feedforward means that the resulting state is in the +1 eigenspace of $S^Z$. This will then be the expected outcome for the value of final $S^Z$ measurement.

For concreteness, we will consider the measurement of logical $ZZ$, for which the final $S^Z$ measurement is achieved through the final measurement of data qubits. The circuit, then, has eight flag results in addition to the above three conditions for syndrome measurements. This gives 11 error-sensitive events in all. To analyse how errors in the circuit are detected, we consider all the possible ways in which Pauli errors can be inserted around each gate. Specifically, we consider the insertion of $X$, $Y$ and $Z$ before any single-qubit gate, and all possible single- and two-qubit Paulis before any two-qubit gate, on the qubits that support the gate. We then simulate each of these circuits to determine how the error is detected.

This analysis has two important uses. First, it can be used to verify the fault tolerance of the scheme, by confirming that all Pauli errors with non-trivial effect are in some way detected by the error-sensitive events. Second, it can be used to determine the specific combination of error-sensitive events, $s$, that detect each error. This information can then be used to infer the corresponding probabilities $\varepsilon_s$ that such errors occurred, by looking at how often the corresponding error signature occurs within the outcomes measured.

After performing this analysis, it was found that the circuit is fault-tolerant. The only cases in which an error was not detected are those where the system was in an eigenstate of the error operator, or where its application was immediately followed by a measurement in an

eigenbasis of the Pauli error. In both of these cases, the error will have a trivial effect on the circuit output.

When calculating the $\varepsilon_s$, it is important to note that the error signatures, $s$, are not necessarily unique for each type of error. For example, $X$ and $Y$ Paulis inserted immediately before any measurement will yield the identical effect of a measurement error. We, therefore, also determine the degeneracy, $N_s$, for each error signature. This is the number of unique errors that gives rise to the same error signature. With this information, we can then analyse the syndrome outcomes from experimental data, looking for these signatures and determining the probabilities with which they occur[32].

Owing to the limited number of error-sensitive events used in this experiment, these probabilities can be calculated directly. The combined probability, $\varepsilon_s$, for all forms of error that lead to a particular signature is determined using the number of shots for which that signature occurs, $n_s$, and the number of shots for which no error is detected, $n_0$. The ratio of these numbers of shots will be the ratio of the probability that the error occurs with the probability that it does not:

$$\frac{n_s}{n_0} \approx \frac{\varepsilon_s}{1-\varepsilon_s}. \tag{9}$$

Simply rearranging this relation gives us the value of $\varepsilon_s$ (ref. 62). We then use the degeneracy to obtain the average probability for each possible single-qubit Pauli error with this signature: $\varepsilon_s/N_s$.

## Standard magic-state preparation circuits

Here we describe magic-state preparation circuits with no error suppression that are compared with our error-suppressed scheme described in the main text.

In Extended Data Fig. 8a, we show a circuit that prepares an encoded CZ state by, first, preparing a CZ state on two physical qubits and, then, encoding the state such that the Pauli observables of the two qubits of the CZ state can be represented as logical operators of the error-detecting code we encode. Finally, we measure the stabilizer operators of the code to encode the state, assuming we obtain the correct stabilizer measurement outcomes. The circuit used for the preparation step is shown in Extended Data Fig. 8b.

We can make use of the stabilizer operators of the CZ state to simplify the preparation circuit shown in Extended Data Fig. 8b. We define a stabilizer operator $U$, with respect to state $|\psi\rangle$, as an operator for which the action is trivial on its respective state, that is, $U|\psi\rangle = |\psi\rangle$. We can check that the CZ state is invariant under the action of a controlled-not gate conditioned on the control qubit in the zero state

$$CX' = |1\rangle\langle 1| \otimes \mathbb{1} + |0\rangle\langle 0| \otimes X.$$

This unitary gate is equivalent to a standard controlled-not gate, $CX = |0\rangle\langle 0| \otimes \mathbb{1} + |1\rangle\langle 1| \otimes X$, followed by a bit flip on the target qubit, that is,

$$CX' = (\mathbb{1} \otimes X)CX.$$

This observation enables us to simplify the preparation circuit. Once the CZ state is prepared, we add the $CX'$ gate in the dashed box in Extended Data Fig. 8, as the state we have prepared at this stage is invariant under this inclusion. The inclusion of this operator enables us to simplify the circuit, as the repeated application of the two Pauli-X rotations and the repeated application of two controlled-not operations used in the circuit act like an identity operation. This trivial step in the circuit is marked on the figure between vertical dashed lines. We can, therefore, omit all of the controlled-not operations and the bit-flip operations from the circuit shown in our implementation of this method of state preparation. As such, this preparation step includes only two entangling gates: a controlled Hadamard gate and a swap

gate. We perform logical tomography by appending the circuits shown in Fig. 2b,c to the end of the circuit shown in Extended Data Fig. 8a. Likewise, we can perform physical tomography on the output of the circuit shown in Extended Data Fig. 8a.

Moreover, we note that the CZ state is also stabilized by the swap gate:

$$\text{swap} = (\mathbb{1} + X \otimes X + Y \otimes Y + Z \otimes Z)/2,$$

and $CZ$ as defined in the main text. The CZ state is uniquely stabilized by the Abelian stabilizer group generated by the set $\langle CZ, CX'\text{swap}\rangle$.

Finally, we also compare our error-suppressed magic-state preparation scheme to a circuit that prepares the same magic state on two physical qubits (Extended Data Fig. 8c). We prepare the state on two physical qubits using a single entangling gate, together with single-qubit rotations, before measuring the state in varying single-qubit Pauli bases, $P$ and $Q$, to conduct state tomography on the circuit output.

## Device overview

Encoded state data collection on ibm_peekskill v.2.4.0 spanned several days over a single region. During this time, monitoring experiments were interleaved with tomography data collection trials. Device coherence times for all qubits exceed about 100 µs and two-qubit errors per gate was found to range from 0.35% to 0.59%. Detailed monitoring of readout errors are provided in Extended Data Fig. 9f,g and time-averaged readout fidelities ranged from 98.1% to 99.6% for all qubits. Average device characterization data are summarized in Extended Data Tables 1 and 2. Unencoded magic-state data were collected over a single 24-h period on ibm_peekskill v.2.5.4 on all physical pairs and the best-performing edge is reported in Extended Data Table 2. Although the unencoded magic-state data were not interleaved with encoded-state tomography, the best-performing pair of physical qubits was found to have a low two-qubit error per gate of 0.38%, and this error is comparable with the lowest two-qubit error per gate for edges used in the encoded magic-state experiments.

## Real-time feedforward control of qubits

In the past decade, several experiments were performed that exploit fast feedback or real-time control within the execution of a quantum program. Fast feedback has been used for conditional reset[63–66], state and gate teleportation[67–69] with low branching complexity and in more demanding algorithms such as the iterative-phase estimation protocol[70], to name a few. More recently, there have been demonstrations of quantum error correction using real-time control in various systems[2,6,71,72]. There have also been examples of work toward classical-control microarchitectures that enable the seamless integration of qubits and classical operations with tens of qubits.

Our work was performed with the first-generation real-time control system of IBM Quantum, in which we use centralized processing of mid-circuit measurement outcomes to classically condition a quantum circuit. The control system architecture is based on a hierarchical heterogeneous system of field-programmable gate array controllers with computing elements for concurrent real-time processing, microwave control and qubit readout. These are synchronized through a global clock and linked with a real-time communication network to enable synchronized collective operations such as control flow. Branching incurs a constant latency penalty to execute the branch (of the order of 500 ns). Real-time computations will incur a variable latency overhead depending on the complexity of the decision. The system provides specialized fast-path control-flow capabilities for rapid and deterministic conditional reset operations. Collective control of the system requires orchestration through a proprietary heterogeneous hardware compiler and code generator. We use an open-access platform that is programmable through Qiskit and OpenQASM 3—an open-source imperative C-style real-time quantum programming

language[73]. All experiments were performed through Qiskit and IBM Quantum Services[74,75].

## Estimates for magic-state yield

Let us attempt to model the error rate of the components of the device using the yield we have evaluated experimentally. The yield is a helpful figure of merit as it tells us precisely how often a single-error event occurs to leading order in the error rate. We first try to model the yield using simple three-parameter models that we derive below. We also compare the yield to numerical simulations of our circuits. We show the estimated yield for different experiments in Extended Data Table 3, in comparison with our analytical model and numerical results.

Both of our analyses have good agreement with the experiment if we assume a two-qubit gate-error rate and a measurement error rate of the order of 2%. This is a high error rate compared with those measured in Extended Data Tables 1 and 2. However, we remark that neither our analytical model nor our simulations account for common error processes such as leakage, cross talk, two-level systems and idling errors that may occur during slow-circuit processes that will introduce additional noise to the system. We suggest discrepancies in our modelling, and the experimentally observed yields can be attributed to these details that are difficult to model analytically or numerically.

Let us present our analytical model to evaluate the yield. We can estimate the magic-state yield as $QR$, where $Q$ is the probability that the random measurement outcomes we obtain throughout our experiment yield the values we need to complete the magic-state preparation scheme and $R$ is the probability that the experiment does not experience a single error.

If we have that $\varepsilon_P$ is the probability that a single parity measurement introduces an error and $D$ is the number of parity measurements that are conducted in an experiment, that is, the depth, we can write $R = (1 - \varepsilon_P)^D$, thereby giving the equation

$$\text{logical yield } = Q(1 - \varepsilon_P)^D. \tag{10}$$

We note that $Q$ and $D$ vary for different experiments.

For our rough calculation, we find reasonably good agreement with the experimental data if we take $\varepsilon_P \approx 22\%$. This equates, approximately, to a two-qubit gate-error rate and a measurement error rate of about 2%. Each parity measurement we perform uses eight entangling gates and three mid-circuit measurements. Therefore, neglecting higher-order terms, we obtain the probability that a parity measurement introduces a single error is

$$\varepsilon_P \approx 8\varepsilon_{2Q} + 3\varepsilon_M, \tag{11}$$

where $\varepsilon_{2Q}$ is the two-qubit gate-error rate and $\varepsilon_M$ is the probability of a measurement error. If we set $\varepsilon_{2Q} = \varepsilon_M = 2\%$, we find that $\varepsilon_P = 22\%$.

We also need to predict $Q$ for different experiments. Let us begin with the error-suppressed experiment in which we use feedforward. Here, in the noiseless case, we have one random measurement outcome, in which we initially measure $\overline{W}$. It is readily checked that the probability that we project the $|++\rangle$ state onto the +1 eigenvalue eigenspace of the $CZ$ operator is $Q_{FF} = \langle++|(1 + CZ)|++\rangle/2 = 3/4$. In the case that we do not use feedforward, in addition to obtaining the correct outcome for the $\overline{W}$ measurement, we must also post-select on obtaining the correct outcome of the initial measurement of $S^Z$. We obtain the +1 eigenvalue subspace of this operator with probability 1/2. We, therefore, have $Q_{PS} = 3/4 \times 1/2 = 3/8$. Finally, in the standard preparation procedure, we measure both $S^Z$ and $S^X$, and we require that both give the +1 outcome. Each measurement gives the correct outcome with probability 1/2. We, therefore, have $Q_{STND} = 1/2 \times 1/2 = 1/4$.

Let us comment on the features of this model that agree with the experiment. First of all, we observe that the error-suppressed scheme using feedforward has a consistently better yield than the other two

schemes, both the error-suppressed scheme using post-selection and the standard preparation scheme. Furthermore, we observe that the error-suppressed post-selection scheme and the standard scheme have comparable yields, for both tomography circuits shown in Fig. 2.

Furthermore, our model explains the difference in yield between different tomography experiments conducted using the two different circuits shown in Fig. 2. The tomography circuit in Fig. 2c uses two additional parity measurements than that shown in Fig. 2b. As such the tomography circuit in Fig. 2c is inherently more noisy than that in Fig. 2b. This is reflected in Fig. 4 in which the yield for tomography circuits shown in Fig. 2b,c are shown in Fig. 4 (left, right).

Our rudimentary analytical model correctly predicts several qualitative features of our experimental data. However, it neglects many details of the circuit. As we might expect, we find better agreement with the experimentally observed yield if we simulate our circuit. We assume an error rate for each of the two-qubit entangling gates and a measurement error rate of 2%. These results are also shown in Extended Data Table 3. Again, the physical error rate of these circuit elements is considerably higher than the observed error rates of these components. As mentioned at the beginning of this section, we attribute this to noise processes that are not captured by either our analytical model or our numerical simulations. In practice, it is extremely difficult to capture all of the physical details that occur in an experiment.

## State tomography with readout error mitigation using noisy positive-operator-valued measurements

The state tomography in the main text uses the Qiskit Experiments implementation of state tomography[31]. A notable change from the previous works is that we do not use readout error mitigation in the main text. Instead, we perform tomographic fitting assuming ideal measurements, which attributes any undetectable measurement errors to errors in the reconstructed quantum state. For physical tomography, we use the cvxpy_gaussian_lstsq fitter with measurement data using the default Pauli-measurement basis on each physical qubit to obtain a weighted maximum-likelihood estimate, constrained to the space of positive, semi-definite, unit trace density matrices. For logical tomography, we use the cvxpy_linea_lstsq fitter with a custom measurement basis using Pauli expectation values, rather than Pauli eigenstate probabilities. In this case, the custom fitter weights are calculated from the inverse of the standard error in the Pauli expectation value estimates for each post-selected logical Pauli operator measurement.

Susceptibility to measurement error is a common issue in tomographic methods. In general, tomographic tools are only as good as the noise model of the measurement apparatus, that is, our ability to calculate the likelihood representing the conditional probability of obtaining a dataset given some test density matrix. In this section, we discuss an alternative approach combining readout error characterization with tomographic reconstruction. Although the dominant measurement error source in tomography experiments is because of qubit readout, it is a common practice to assume local, uncorrelated readout errors in the $Z$ basis. A set of noisy positive operator-valued measurements (POVMs) on a single-qubit is,

$$Z_0' := \begin{bmatrix} 1-p & 0 \\ 0 & q \end{bmatrix}, Z_1' := \begin{bmatrix} p & 0 \\ 0 & 1-q \end{bmatrix}, \tag{12}$$

where $p$ is the probability of assigning outcome 1 to a state initially prepared as $|0\rangle$ and $q$ is the probability of assigning outcome 0 to a state initially prepared as $|1\rangle$; that is, $p = P(1|0)$ and $q = P(0|1)$. We can also construct noisy POVMs for measurements in the Pauli-X or Pauli-Y eigenbases by rotating the noisy POVMs shown in equation (12) by an appropriate angle assuming ideal unitaries, because the measurement error is typically several orders of magnitude greater than the one-qubit gate error.

By interleaving small batches of experimental data collection with readout calibration experiments, one can construct noisy POVMs for each data qubit applicable to a small duration of data collection to be used in fitting procedures discussed above. In Extended Data Fig. 9a, state infidelities from fitting with noisy POVMs can be compared with fitting with ideal projectors ($p, q \equiv 0$), in which the latter is reported in the main text. Using readout mitigation, the fault-tolerant tomography routines far outperform both unencoded tomography and the physical tomography of the encoded state. As the terminating measurements in logical tomography are very similar to those in physical tomography, we would expect both of these experiments to demonstrate similar infidelities. Resolving this discrepancy remains an open research question.

Furthermore, it is unclear if our assumed construction of noisy POVMs, or the measured readout error calibrations, collectively reflect the true measurement errors experienced by data qubits. We, therefore, test the sensitivity of the outcomes of state tomography to the choice of measurement compensation in Extended Data Fig. 9b–d. State infidelity is calculated by fitting experimental tomography data to POVMs parameterized by $p$ and $q$. To simplify, these readout error probabilities are set to be constant for all qubits and time. Dark-blue regions of low infidelity (with the minima marked with a red star) do not coincide with the state infidelity calculated using the global average of experimentally measured readout calibrations (marked by a black dot). This disparity suggests that either the target experiments experienced initialization or measurement errors at a higher rate than measured by simpler calibrations and/or fitting with potentially incorrect $A$-matrices yields a highly non-positive state that is mapped to a high-fidelity physical state under constrained optimization.

Combining readout mitigation with tomography thus remains an open question for further work, and the results of the main text are limited by unaddressed readout error on terminal measurements. We expect that state tomography experiments in Extended Data Fig. 9b–e at $p = q = 0$ provide a reasonable upper bound on the error of the underlying magic state.

## Data availability

The datasets generated and analysed during this study are available at https://doi.org/10.6084/m9.figshare.23535237.

## Code availability

The codebase used for data analysis and figure generation is available at https://doi.org/10.6084/m9.figshare.23535237; other supporting codes are available upon reasonable request.

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

**Acknowledgements** We acknowledge the use of IBM Quantum services for this work. These system capabilities are available as open access to device users. We also acknowledge the work of the IBM Quantum software and hardware teams that enabled this project. The views expressed are ours and do not reflect the official policy or position of IBM or the IBM Quantum team. B.J.B. is grateful for the hospitality of the Center for Quantum Devices at the University of Copenhagen. J.R.W. acknowledges support from the NCCR SPIN, a National Centre of Competence in Research, funded by the Swiss National Science Foundation (grant no. 51NF40-180604). R.S.G. and S.T.M. acknowledge support from the Army Research Office under QCISS (W911NF-21-1-0002). T.A., M.H., and M.B.H. acknowledge support from IARPA under LogiQ (contract W911NF-16-1-0114) on real-time control software work. All statements of fact, opinion or conclusions contained herein are ours and should not be construed as representing the official views or policies of the US government.

**Author contributions** R.S.G., N.S., T.A., M.H. and M.B.H. enabled experimental execution using real-time control flow; S.T.M. performed unencoded magic-state tomography experiments; R.S.G. performed encoded magic-state tomography experiments and simulations; C.J.W. and S.T.M. conceived and developed tomographic fitting procedures, with and without error mitigation, implemented by R.S.G; J.R.W. conducted numerical simulations to test the fault-tolerant properties of the preparation circuits and for analysis of the experimental results; R.S.G., C.J.W., S.T.M., J.R.W., M.T. and B.J.B. performed data analysis; T.J.-O., T.J.Y., A.W.C. and B.J.B. developed the fault-tolerant magic-state preparation circuits; R.S.G. assumed primary responsibility for experimental execution, analysis, codebase and data management; M.T. and B.J.B supervised the project; R.S.G., N.S., T.A., C.J.W., S.T.M., J.R.W., M.T. and B.J.B. wrote the paper with input from all the authors.

**Competing interests** A patent (application no. 18/053087) was filed on 7 November 2022 with listed inventors B.J.B., A.W.C., R.S.G., T.J.-O. and T.J.Y. The authors declare no other competing financial or non-financial interests.

**Additional information**
**Correspondence and requests for materials** should be addressed to Benjamin J. Brown.

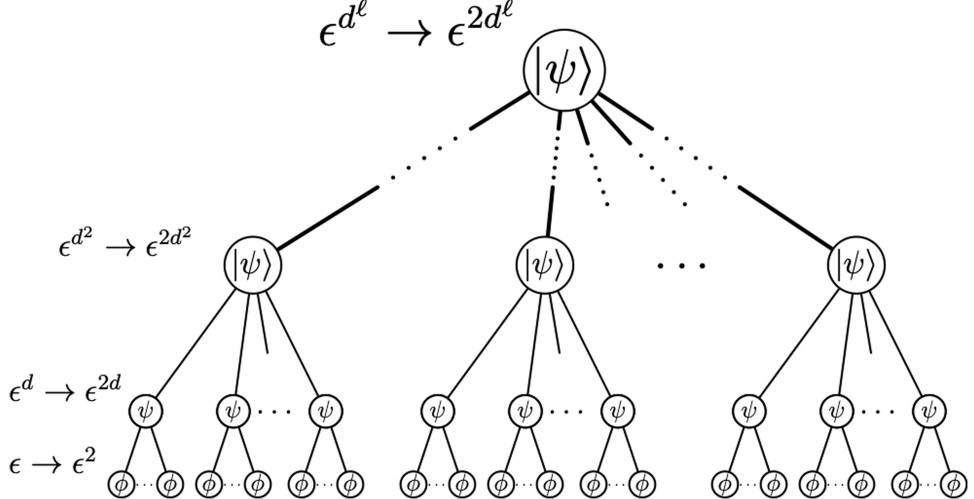

**Extended Data Fig. 1 | A generic magic-state distillation protocol.** Encoded input magic states are combined such that higher fidelity magic states are produced with some probability. For a single use of a magic-state distillation protocol, the error of an input magic state $\epsilon$ is suppressed like $\epsilon \to \epsilon^d$ where $d$ is a constant determined by the magic-state distillation protocol. Applying distillation recursively allows us to produce magic states with an arbitrarily high fidelity. By initializing error-suppressed magic states in the first step, where the error is suppressed as $\epsilon^2$ we obtain a quadratic improvement in the fidelity of the output magic state.

(left)                                    (right)

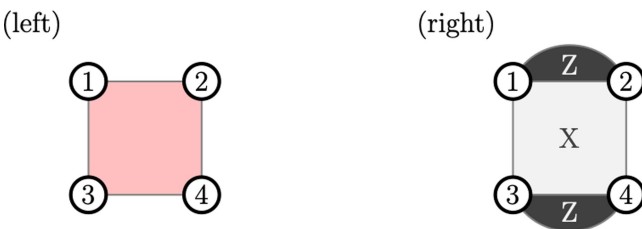

**Extended Data Fig. 2 | Small codes.** We describe how to encode these codes into higher distance codes. (left) The error-detecting code prepared in the main text. We refer to this code as the [[4, 2, 2]] code to distinguish it from the [[4, 1, 2]] code shown to the right of the figure. The [[4, 2, 2]] code has stabilizer generators $S^X = X_1X_2X_3X_4$ and $S^Z = Z_1Z_2Z_3Z_4$ and logical operators $\overline{X}_A = X_1X_2$, $\overline{Z}_A = Z_1Z_3$, $\overline{X}_B = X_1X_3$ and $\overline{Z}_B = Z_1Z_2$ for logical operators indexed $A$ and $B$. (right). The [[4, 1, 2]] code is an error detecting code that encodes a single logical qubit. It is closely related to the error detecting code shown (left). It has stabilizer generators $S^X = X_1X_2X_3X_4$, $S_T^Z = Z_1Z_2$ and $S_B^Z = Z_3Z_4$, and logical operators $\overline{X} = X_1X_2$, $\overline{Z} = Z_1Z_3$.

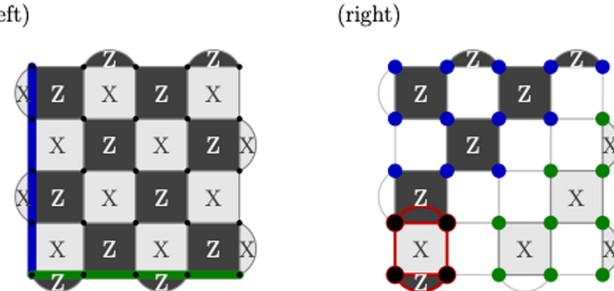

**Extended Data Fig. 3 | Injecting an encoded magic state into the surface code.** The magic state is initially encoded on a [[4, 1, 2]] code. (left) The standard surface code with physical qubits on the vertices of a square lattice and standard Pauli-X and Pauli-Z type stabilizers marked by lattice faces. Supports for the logical Pauli-X and Pauli-Z operators are shown in green and blue, respectively. (right) We show the initial state that is injected into the surface code. The [[4, 1, 2]] code is shown in red in the bottom-left corner. The remaining qubits of the surface code lattice are prepared in a product state, where blue (green) qubits are prepared in the $|0\rangle_v$ ($|+\rangle_v$) state. We show the code deformation stabilizers, i.e. $\mathcal{S}_{\text{def.}} = \mathcal{S}_{\text{init.}} \cap \mathcal{S}_{\text{fin.}}$, shaded on the right lattice.

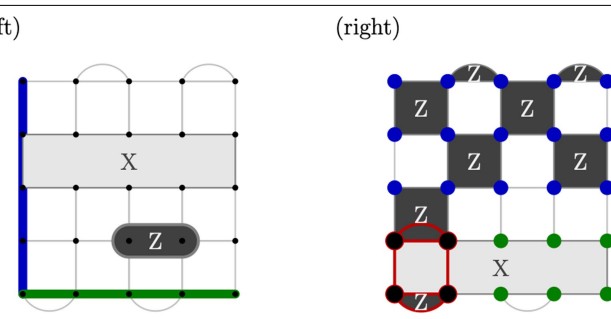

**Extended Data Fig. 4 | Injecting an encoded state into the heavy-hex code.**
The injected state is initially encoded on the [[4, 1, 2]] code. (left) A lattice with
qubits on the vertices. We show the support of a single Pauli-Z gauge check and
a Pauli-X stabilizer operator. The support of the Pauli-Z gauge check is shown
in dark gray. The Pauli-X stabilizer operator is shaded grey towards the top of
the lattice. We also show the support of a Pauli-X- and Pauli-Z-type stabilizer in
green and blue, respectively. (right) The stabilizer group for $\mathcal{S}_{\text{init.}}$. The [[4, 1, 2]]
code is outlined in red in the bottom-left corner of the lattice. The other qubits
are initialized in a product state with blue (green) qubits initialized in the $|0\rangle$
($|+\rangle$) state. Stabilizer operators $\mathcal{S}_{\text{def.}} = \mathcal{S}_{\text{init.}} \cap \mathcal{S}_{\text{fin.}}$ are shaded in the figure.

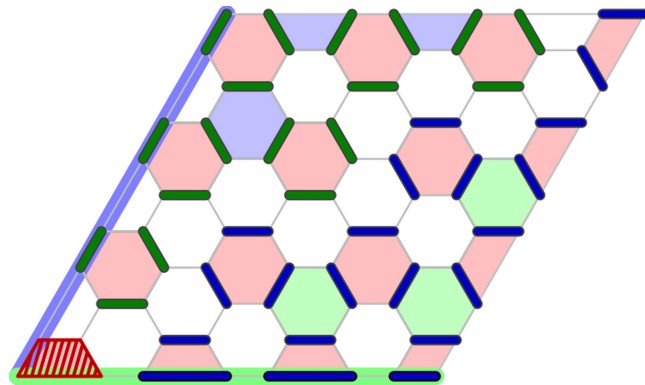

**Extended Data Fig. 5 | Injecting an encoded two-qubit state into the color code.** The state is initially encoded with the [[4, 2, 2]] code. A qubit is supported on each of the vertices of the lattice. We initialize the system $\mathcal{S}_{\text{init.}}$ such that the [[4, 2, 2]] code, shaded in red, is supported on a weight-four face in the bottom left corner of the lattice. The other qubits are prepared in Bell pairs on the highlighted blue and green edge terms. As such, we shade the faces of $\mathcal{S}_{\text{def.}}$ where both a Pauli-X and Pauli-Z stabilizer is supported. The support of the logical operators on the left and bottom boundaries are highlighted in blue and green, respectively.

### 1. Prepare

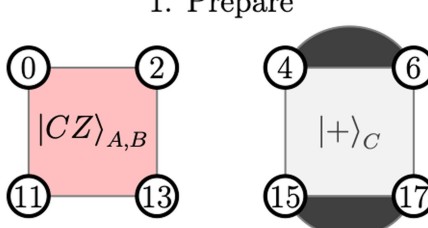

### 2. Transport

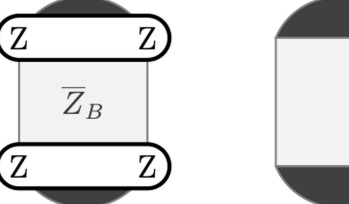

### 3. Logical parity measurement

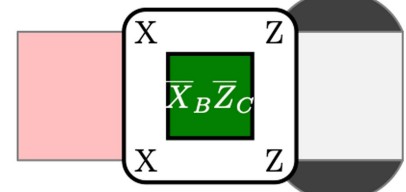

### 4. Logical measurement

**Extended Data Fig. 6 | Preparing a CZ state over two [[4, 1, 2]]-codes.** At step 1 the codes are prepared. The [[4, 2, 2]] code that encodes the two-qubit CZ-state is represented by the red square where its four qubits lie at the vertices of the square. This preparation is described in the main text. The code is prepared adjacent to a [[4, 1, 2]]-code that is initialized in an eigenstate of the $|+\rangle$ state. The qubits in the figure are indexed according to the qubit-map shown in Extended Data Fig. 7. At step 2 the qubits are transported in order to perform a logical parity measurement in step 3 using the heavy-hex lattice geometry. Note that the qubit indices have changed. This step can be performed with swaps, for instance, as shown in Extended Data Fig. 7(top). At step 3 a logical parity measurement is made. It can be performed in a fault-tolerant manner using qubits 5, 10, and 16, as shown in the green box in Extended Data Fig. 7 (bottom). We complete the operation by measuring the logical operator $\overline{Z}_2$ in step 4. This weight-two measurement can be repeated in two locations on the [[4, 2, 2]] code such that a single measurement error can be detected. This final measurement projects the [[4, 2, 2]] code onto the [[4, 1, 2]]-code by reassigning the $\overline{Z}_B$ logical operators as stabilizers of the system.

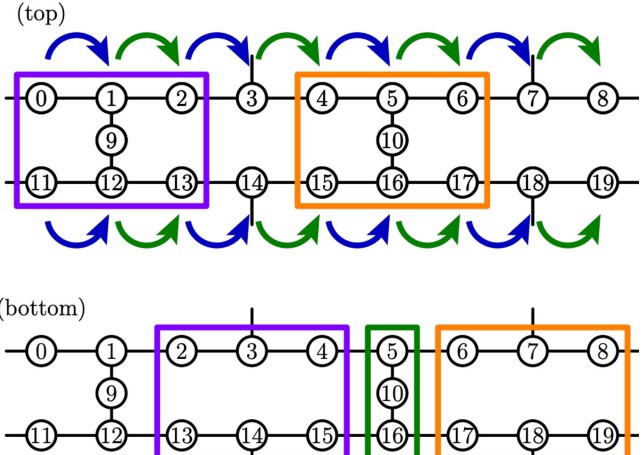

(top)

(bottom)

**Extended Data Fig. 7 | Mapping the encoding onto the heavy-hexagonal lattice geometry.** We encode the CZ-state onto two copies of the [[4, 1, 2]]-code. (top) We prepare the encoded CZ-state as defined in the main text using the qubits outlined in the purple box. We additionally prepare a [[4, 1, 2]]-code in the logical |+⟩ state using the qubits outlined in the orange box. To perform step 3, as shown in Extended Data Fig. 6, we first move the codes, as in step 2. This can be performed using swap gates between adjacent qubits. Swap gates are performed, first, between pairs of qubits marked by a blue arrow, and then between pairs of qubits marked with green arrows. Each set of swap gates, the blue set and the green set, can be performed in parallel. Completing the swap operations moves the codes over the qubit map. We show the locations of the codes after the swap operations by outlining their supporting qubits with a purple and orange box, respectively, in the bottom figure. In their new locations, the logical parity measurement of step 3 can be performed using ancillary qubits 5,10 and 16, outlined in the green box in the bottom figure. At the final step we facilitate the measurement of $Z_4Z_6$ and $Z_{15}Z_{17}$ using ancillary qubits 5 and 16, respectively.

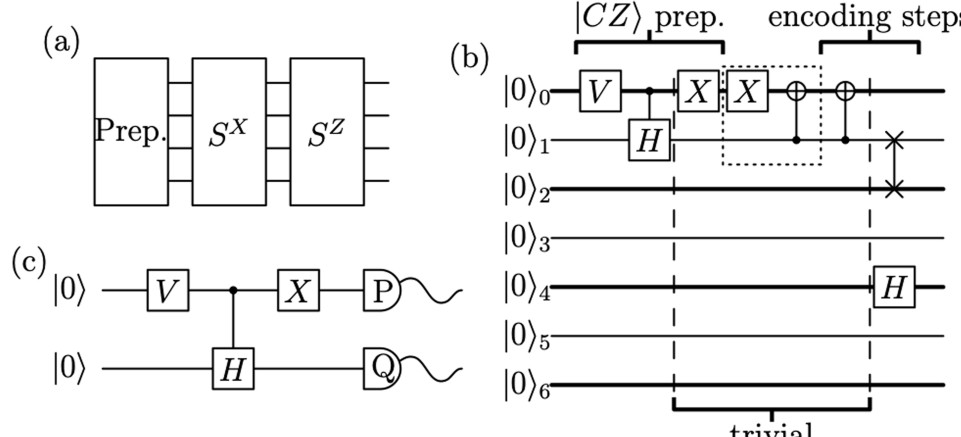

**Extended Data Fig. 8 | Magic-state preparation without error suppression.** We can encode a physical CZ state using the circuit outlined in (a), where the preparation step, Prep., is shown in (b). The magic state is then encoded using stabilizer measurements $S^x$ and $S^z$. The preparation circuit, (b), first prepares a CZ state on two physical qubits before preparing the state to encode it in the four-qubit code by stabilizer measurements. The circuit makes use of $V = \exp(i\theta Y)$ a Pauli-Y rotation with $\tan\theta = \sqrt{2}$, a controlled-Hadamard gate and a bitflip. We find that we can simplify the circuit once the CZ state is prepared by making use of the stabilizer operators of the CZ state. As discussed in the main text we observe that the circuit element in the box with a dotted outline acts trivially on the CZ-state. The inclusion of this stabilizer operator allows us to remove all of the Pauli-X and controlled-not operations shown in the circuit, as the circuit elements in the box negate their adjacent self-inverse gates. Indeed, the circuit elements that lie in between the vertical dashed lines act like the identity operator. (c) The CZ state is prepared on two physical qubits, where the circuit elements are defined above. We perform state tomography on this state by making different choices of single-qubit Pauli measurements, $P$ and $Q$, on the output of this circuit.

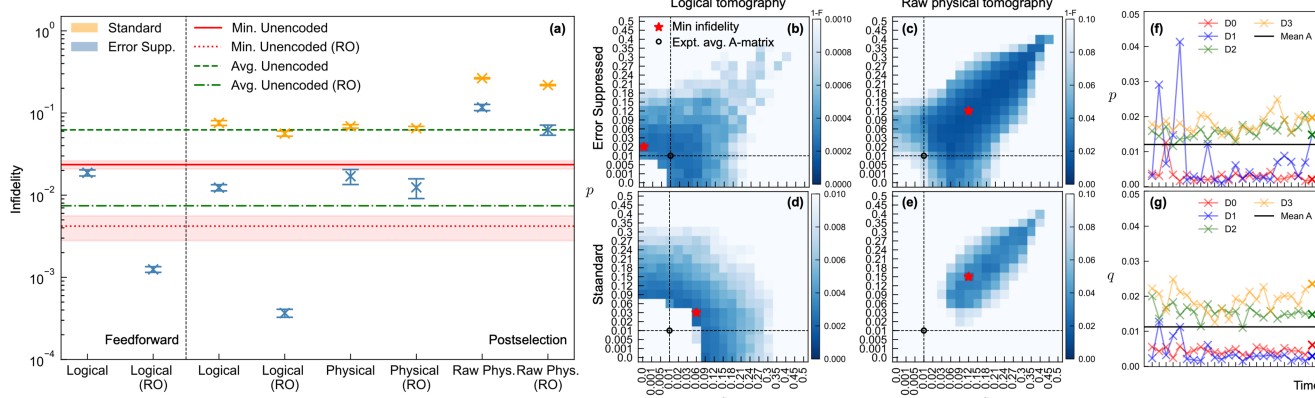

**Extended Data Fig. 9 | Combining readout-error mitigation with state tomography methods.** (a) State infidelity for the standard (orange) vs. error-suppressed (blue) schemes using different tomographic methods; error-bars represent 1σ std. dev. from bootstrapping. On the x-axis, a state is reconstructed with either logical tomography (Logical) or physical tomography after logical projection (Physical); tomography assumes either ideal projectors, as in the main text, or noisy POVMs representing uncorrelated, local readout errors (RO) on terminal data qubit measurements. Raw physical tomography (Raw Phys.) refers to the state on four physical qubits prior to logical projection. Red dotted (green dot-dashed) lines show lowest (average) state infidelities of the two-qubit unencoded magic state prepared with RO mitigation. With RO mitigation, logical tomography outperforms the min. unencoded state supporting conclusions in the main text. (b)-(e) Heatmap of state infidelity vs. avg. measurement error, $p \equiv P(1|0)$, $q \equiv P(0|1)$. Experimental tomography data is fit to noisy POVMs using a parameterized $A$-matrix, $A := [[1 - p, q], [p, 1 - q]]$, where $p$, $q$ are constant for all qubits and time. Experimental readout calibrations data are averaged over time and qubits, and correspond to a single state infidelity in (b)-(e) (black dots). These state infidelities (black dots) do not coincide with local minima (red stars) or even high-fidelity regions. (f)-(g) Readout calibration measurements of $p$, $q$ vs. time for all four data qubits over several days; average rates (black solid) are used in (b)-(e) for state fidelities marked by black dots.

**Extended Data Table 1 | Average single-qubit benchmarks**

| Qubit ($Q_F$) | Freq. (GHz) | Anharm. (MHz) | $T_1$ ($\mu$s) | $T_2^{echo}$ ($\mu$s) | EPG (%) | Readout Fid. (%) | $P(0\|1)$ | $P(1\|0)$ |
|---|---|---|---|---|---|---|---|---|
| 17 | 5.151 | -339.9 | 256.3 | 170.7 | 0.024 | 99.6 | 0.00472 | 0.00305 |
| 18 | 5.083 | -341.9 | 182.2 | 364.9 | 0.024 | 98.7 | 0.01297 | 0.01240 |
| 21 | 4.858 | -344.2 | 366.7 | 362.2 | 0.012 | 99.4 | 0.00318 | 0.00822 |
| 15 | 4.958 | -343.8 | 212.4 | 200.5 | 0.036 | 98.5 | 0.01658 | 0.01313 |
| 10 | 4.837 | -345.0 | 200.6 | 120.9 | 0.027 | 98.4 | 0.01612 | 0.01542 |
| 12 | 4.899 | -346.8 | 289.7 | 462.6 | 0.036 | 98.7 | 0.01505 | 0.01068 |
| 13 | 4.972 | -345.8 | 322.4 | 166.1 | 0.024 | 98.1 | 0.01952 | 0.01818 |

Data shown is for qubits of ibm_peekskill used in this work.

**Extended Data Table 2 | Average two-qubit gate benchmarks**

| Gate | CX length (ns) | EPG (%) |
|---|---|---|
| 12_10 | 334.2 | 0.58 |
| 15_12 | 376.9 | 0.59 |
| 13_12 | 462.2 | 0.37 |
| 15_18 | 376.9 | 0.56 |
| 18_17 | 640.0 | 0.43 |
| 18_21 | 462.2 | 0.35 |
| 18_21* | 462.2 | 0.38 |

Data shown are for the qubits of ibm_peekskill used in this work. CX gates, constructed from echoed cross-resonance pulse sequences, are specified in one direction, with the reverse directions accessed by the addition of single-qubit gates. Error per gate (EPG) is extracted from isolated two-qubit randomized benchmarking (spectator qubits idling). The notation * denotes error rates for the best performing physical qubit pair on ibm_peekskill during unencoded magic state preparation experiments defining the minimum (red line) in Fig. 3.

**Extended Data Table 3 | Estimated magic-state yield compared with experiment**

| circuit | $Q$ | $D$ | analytic | numeric | experiment |
|---|---|---|---|---|---|
| FF 2(b) | 3/4 | 4 | $\sim 28\%$ | $\sim 35\%$ | $\sim 30 - 35\%$ |
| FF 2(c) | 3/4 | 6 | $\sim 17\%$ | $\sim 14\%$ | $\sim 10\%$ |
| PS 2(b) | 3/8 | 4 | $\sim 14\%$ | $\sim 20\%$ | $\sim 17\%$ |
| PS 2(c) | 3/8 | 6 | $\sim 9\%$ | $\sim 8\%$ | $\sim 5 - 8\%$ |
| standard 2(b) | 1/4 | 3 | $\sim 12\%$ | — | $\sim 15 - 17\%$ |
| standard 2(c) | 1/4 | 5 | $\sim 7\%$ | — | $\sim 4 - 6\%$ |

We compare our analytical model, Eqn. (10), and numerics to the experimental data. We calculate the yield for the error-suppressed preparation experiment using feedforward (FF) and the error-suppressed preparation experiment using (PS). We also estimate acceptance rates for the standard experiment. The depth of the circuits $D$ vary depending on the different tomography experiment we run, so we treat them separately. We append 2(b) and 2(c) to the different experiments depending on the tomography circuit we used, in reference to the circuits shown in Fig. 2(b) and (c) in the main text.