## [Peer Review File · Nature]

Manuscript Title: Encoding a magic state with beyond break-even fidelity

Reviewer Comments & Author Rebuttals

Reviewer Reports on the Initial Version:

Referees' comments:

Referee #1 (Remarks to the Author):

SUMMARY

The authors introduce a fault-tolerant error-detecting state preparation gadget that prepares a particular two-qubit “magic state” encoded in the four-qubit error-detecting code. The circuit involves measuring the eigenvalue of a particular non-Pauli operator \overline{W} , and fault-tolerant error-detecting stabilizer measurement gadgets are derived by substituting \overline{W} with stabilizer generators $S^{X,Z}$.

The authors define two ways to obtain state information after running their circuits, performing either full “physical tomography” on the four data qubits or “logical tomography” via measuring the 15 logical operators of the code.

In a fault-tolerant error-detecting gadget, one post-selects only the runs where no fault is detected. In a step of their state preparation circuit, however, the authors utilize an alternative strategy using an “adaptive circuit element”, i.e., a single-qubit unitary conditional on the outcome of an intermediate measurement. They compare the two strategies experimentally on the IBM `ibm_peekskill` superconducting circuit device, with the adaptive variant yielding lower fidelity but higher magic-state yield than the more standard post-selection variant, presumably due to being able to convert circuit runs that would have otherwise failed under post-selection. This comparison uses logical tomography, and an additional reference fidelity is obtained for the post-selection variant using physical tomography.

To collect additional reference fidelities, the authors experimentally realize logical and physical tomography on an unencoded variant of the circuit that prepares a two physical-qubit (i.e., unencoded) magic state. The fidelity of the encoded variants is higher than the minimal fidelity of the unencoded variant, demonstrating the effectiveness of fault tolerance.

The authors also experiment with new tomographic procedures based on fitting noiseless projective measurements with noisy projection-operator-valued measures (POVM), highlighting several open theoretical and experimental problems.

EVALUATION

This paper presents new theory in the form of several fault-tolerant gadgets and efficient

tomographic procedures for the four-qubit error-detecting code. The key theoretical result seems to be a fault-tolerant way to distill a magic state using four qubits — an elegant circuit that requires only transversal gates to implement. This is a step forward relative to previous work, which only realized Clifford-group gadgets and error-correction rounds, but the theory alone does not merit the broadest of audiences, in part because it is not clear how the techniques here can be scaled to larger codes and devices.

The authors also realize said gadgets on a small device, presenting a small realization of another building block of a quantum computer. In terms of required experimental resources, the gadgets seem to be of similar complexity as previous ones and utilize only error detection (as opposed to correction). A purpose for realizing such small gadgets [arXiv:1610.03507] in the past has been to showcase the relative usefulness of and control over devices, but larger and reasonably reliable devices have long been in service since then [see refs. citing arXiv:1610.03507]. It thus seems a bit out-of-date to showcase them in an ultra high-profile publication.

The “adaptive” strategy, highlighted throughout the work, is (to my understanding) a single-qubit Pauli-X operation conditional on a stabilizer measurement being -1 . Such a Pauli can act on any qubit because the goal is to map an all-odd-parity binary string state into an all-even-parity state. This could be a coincidental feature of this combination of code and circuit, but I am open to arguments that this can also be useful for larger CSS codes. I have also not been convinced that a such a seemingly simple operation is a breakthrough experimentally, but I am open to arguments I made have missed.

The above is not meant to take away from the opinion that this is interesting and important work that seems to have been done carefully and that deserves to be published. I appreciate the attention to experimental detail, such as discussion of how non-trace-one experimental “states” are mapped to the closest density matrix, and an attempt to handle noise drift by performing millions of runs over several days. The use of full state tomography instead of readout error mitigation and analysis of the different sets of POVMs should be useful to the field.

OTHER DETAILS

- More work needs to be shown that backs the statement “each [single-qubit Pauli error] will be detected by some combination of the error sensitive events”. I do not think we should require a complete table of all possible faults with an explanation or diagram of how each fault propagates, but a justification that a circuit is fault tolerant should be more than one sentence stating the result.

– In particular, the work could use more detail around the novel part of the magic-state circuit where one is required to be in the $+1$ eigenspace of $S^{\{Z\}}$. Also, one could use more discussion of how the adaptive operation would be compatible with errors that happen before it and after the stabilizer is measured to be -1 .

– For two-qubit gates, single faults can occur on two qubits at the same time, and a justification as to how the gadgets can tolerate them is in order alongside the single-qubit discussion in, e.g., Appx. E.

- There are a few typos:

- “than they physical state”

- Fig. 4 caption: “Yield or acceptance rate after post-selection (%) vs. Logical tomography circuits from”. “vs” should be “for”?

- “state is prepared is”

Referee #2 (Remarks to the Author):

One pathway to reach universal error-corrected logical quantum computation is the preparation of high-quality magic states, for the realisation of non-Clifford quantum gate operations by gate injection. Towards addressing this key challenge, the present work represents an important step forward and contains the following key results:

A new fault-tolerant protocol to encode two-qubit magic states, so-called CZ-states, encoded in a 4-qubit quantum error detecting code. Whereas this $[[4,2,2]]$ code is well known and has been implemented before, the key new insight is that, in the spirit of flag-qubit based fault-tolerance, a fault-tolerant gate sequence can be designed that effectively measures the logical controlled-Z operator on the two encoded qubits.

Experimentally, the work demonstrates an implementation of the encoding in a superconducting quantum processor with qubits arranged according to the heavy-hexagon layout. Here, the key result is that the work achieves lower error rates of fault-tolerantly encoded magic states than for the unencoded respective states on two physical qubits. Furthermore, the work uses real-time in-sequence feed-forward under which the fidelity of the prepared magic-states remains superior to the unencoded ones, while the gain increases.

Reaching this break-even point, to the best of my knowledge for the first time, for an encoded magic state, is an important milestone and conceptual demonstration towards universal fault-tolerant quantum computing. I find the results, in particular the FT preparation protocol, very original and think that they will be of immediate interest for the field of quantum error correction and quantum fault-tolerance, and could, in principle, warrant publication in Nature.

However, I ask the authors to address the following points in a revision:

- 1) The manuscript is overall clearly written and can be followed well. However, the explanation of the context of the experiment should be strengthened. This is needed to make the relevance clearer to non-expert readers, who might not be experts in the variants of magic-state preparation and gate injection. Here, in the current version, the motivation for the proposed protocol is summarised only in a single sentence ‘[...] CZ states can be used to prepare Toffoli states [23] that can be distilled [15].’ (note also the typo). This should be elaborated on, ideally in the main text, at least outlining briefly how the 3-qubit Toffoli states (non-encoded) can be obtained from the CZ-states prepared in

this work, and how such Toffoli-states enable universal logical quantum computation (in combination with the Hadamard). To this end, alternatively, an additional background methods section covering this less well known route could be appropriate.

2) The work comprehensively analyses the experimental measurement data by various tomographic techniques, which are clearly explained. It would, however, in my opinion also be important to understand to which extent the experimental results can be understood or reproduced from numerical simulations of the noisy quantum circuits, e.g. based on a noise model extracted from the experimental error rates of the devices components. In particular, how can one understand the quite strongly varying acceptance probabilities in the various logical bases? (see Fig. 4, from 15-30% to about 5%)?

3) Along a similar line: based e.g. on numerical simulations, can the authors comment on what the main bottleneck of the proposed protocol is? For example, it would be interesting to understand what the expected improvements over the observed improvement of about 4 would be as gate error rates and / or qubit measurement rates are changed. What are the expected performance gains from improving on these parameters in future, improved hardware?

4) A brief outlook of the work, possibly in the form of an additional methods / background section, would be important, again to highlight the context and what doors the present work opens. The current outlook sentence In the future it will be exciting to continue to design, develop and test new gadgets with real hardware, that will improve the performance of the key subroutines needed for fault-tolerant quantum computing. is very generic. Specifically, what are next or potential future steps that this specific demonstrated protocol opens? For example, can it be scaled up and how? Can this approach be generalised beyond the demonstrated fault-tolerant encoding in an error detecting code to error-correcting, possibly larger-distance codes? What kind of architectural constraints does this pose for the required quantum hardware? E.g. could the generation of Toffoli-states (and subsequent distillation of the latter) from these encoded CZ-states be achieved in a larger heavy-hexagon quantum processor architecture or what architectural adjustments or capabilities would be generally required?

Referee #3 (Remarks to the Author):

The manuscript "Encoding a magic state with beyond break-even fidelity", by Gupta et al, presents theoretical analysis and experimental results for several protocols distilling a magic state, including a novel fault-tolerant protocol based on a 4-qubit code with flag-qubit measurements. Generally, magic states are an important resource for fault-tolerant quantum computation beyond quantum memory. Such states are used to perform non-Clifford gates; an ability to prepare and distill magic state efficiently and with high fidelity is an important milestone for universal scalable quantum computation. While the magic state distillation fidelities in the present

experiments are not that high (just under 2%), an impressive result is that the fault-tolerant protocol gives better fidelity than would be possible with a very short conventional unprotected circuit on any pair of qubits used in this experiment. This is despite that fact that 7 qubits are used with a high-depth circuit.

In my opinion, this manuscript presents an important and exciting experimental result, and I strongly recommend it for publication. I do have several questions and comments to the authors.

1. What are the expected leakage rates on the devices and gates used in the experiment? Is there any quantitative data about leakage rates? Assuming leakage rates around 0.1% per gate or smaller, with the depth of the circuits used, accumulated leakage should not be a problem in a single round. However, is there enough wait time between subsequent rounds of experiment to ensure small enough probability of leakage to be present at the start?
2. Do I understand correctly that in the post-selection version of the experiment, instead of the classical line on Fig. 2(a), the experiment is just stopped after an incorrect measurement result? Is the difference between the FF and PS state fidelities in Fig. 3 consistent with the increased experiment time, or are there additional error channels associated with the feedforward?
3. What is the function of the H gate on line 4 in Fig. 5(b)? In general, I find the discussion about "removing all of the Pauli-X and controlled-not operations" a bit unclear; perhaps it would be better to give the original version of the circuit with these operations included?
4. Minor corrections: page 3, left column, end of the 3rd line in the 1st complete paragraph, replace "rather they" with "rather than".
5. In Fig 2 caption, I would replace unconventional "stabilizer data" with a more commonly used "syndrome" or "syndrome bits".
6. I believe the term "stabilizers" (albeit its common use) to be a bit colloquial, and suggest to use "stabilizer generators" instead (page 1, last paragraph on the right).

Author Rebuttals to Initial Comments:

In response to referee #1:

Many thanks for identifying the importance of our work, and for your constructive feedback. We comment on some of the remarks in your evaluation below, and detail how we have amended our manuscript in response to these remarks.

“... the theory alone does not merit the broadest of audiences, in part because it is not clear how the techniques here can be scaled to larger codes and devices.”

We demonstrate a component that can be used in magic-state distillation protocols in large-scale fault-tolerant quantum-computing architectures in the future. Magic-state distillation is a well-developed method to complete a universal gate set with many practical quantum computing architectures. Essential to magic state distillation is the preparation and injection of a magic state into larger quantum error-correcting codes. Preparing better magic states on small codes, as we have demonstrated in our break-even experiment, helps us inject better input magic states into state distillation circuits, which will ultimately lead to reductions in the resource cost of distilling high-fidelity magic states in large-scale quantum computers. In this sense, the path to using our component in a scalable system is clear.

On this point, we believe that readers may benefit from additional details on the wider context of our experiment with respect to magic-state distillation and state injection than those we briefly presented in our original submission.

In response to this comment we have written a supplementary document explaining how the prototype we have demonstrated can be used in magic-state distillation protocols run on large-scale devices in the future. This includes details for how we might inject the magic state we have prepared into the heavy-hex code prepared on the device used to conduct the experiment.

“... larger and reasonably reliable devices have long been in service since then [see refs. citing arXiv:1610.03507]. It thus seems a bit out-of-date to showcase [small gadgets] in an ultra high-profile publication.”

We must overcome many challenges to produce a large-scale quantum computer. These include demonstrations of both larger devices as well as improving the performance of computational operations using these devices. By outperforming the physical operations of our device, our break-even result is a milestone demonstrating the latter. Indeed, our break-even result demonstrates unprecedented control over superconducting qubits to perform essential protocols for fault-tolerant quantum computing. We remark that the use of quantum error correction has enabled this result.

Given a device with more qubits that is capable of large-scale quantum computing via the distillation of magic states, it would still be valuable to implement the same seven qubit experiment, as we may then continue to inject the state into a larger code. We have described such injection operations in the Supplementary Information in our revised submission. As such, it is important to test and verify the performance of small components used in any such system. The improved fidelity we have demonstrated with our break-even result indicates that the new scheme we have proposed may indeed reduce the resource overhead of magic state distillation in large-scale fault-tolerant systems.

‘I am open to arguments that [the adaptive strategy] can also be useful for larger CSS codes.’ I have also not been convinced that a such a seemingly simple operation is a breakthrough experimentally, but I am open to arguments I made have missed.’

Adaptive circuits where circuits are updated depending on the outcome of mid-circuit measurements are a key component of all practical proposals for large-scale fault-tolerant quantum computing. As a simple yet essential example, logic gates that are performed by state teleportation will require a logical circuit that is adapted depending on the outcome of a fault-tolerant measurement. Likewise, the yield of magic-state distillation circuits will benefit greatly from using adaptive circuits to apply a Clifford correction operator to the output state. Other models for fault-tolerant quantum computing that complete a universal gate set using, say, gauge fixing or just-in-time decoding will also demand circuit updates according to the outcomes of random measurements. In fact, given that universal fault-tolerant quantum computing is impossible using a transversal (unitary) set of logic gates due to the Eastin Knill theorem, it seems unlikely that one could conceive of a practical model for fault-tolerant quantum computing that does not make use of mid-circuit measurements and feedforward. Overall, with these examples and no-go theorems, it seems a difficult to argue that adaptive circuits will not be vital to a functioning fault-tolerant quantum computer.

Given the importance of adaptive circuits, we must develop hardware that can perform feedforward operations in tandem with the quantum hardware that executes the quantum circuits themselves. This is particularly challenging in superconducting architectures where circuits are updated in response to classical signals, that are produced from measurements made at millikelvin temperatures, that need to be processed much faster than the microsecond clock speed of a superconducting device. In this sense, results demonstrating adaptive circuits in superconducting qubit devices should be celebrated, as this is necessary progress towards large-scale quantum computer. The simple adaptive circuit we present demonstrates the function of the adaptive hardware we use to conduct our experiment. Furthermore, the yield of magic states we produce significantly increases by using this technology. Remarkably, we can perform these adaptive circuits without a significant drop in fidelity, thereby indicating we can perform these adaptive operations very quickly, of the order of hundreds of nanoseconds.

This demonstration represents progress towards the adaptive hardware we will need for large-scale logical fault-tolerant logical operations.

In response to this comment, in order to present the wider context of developing adaptive circuits, we have amended the discussion section to mention where dynamic circuits will be used in large-scale quantum computing architectures. Specifically, we have written:

“The yield of magic states benefited from the use of dynamic circuits where mid-circuit measurements condition gates operations in real time. Remarkably, we find that the operation is sufficiently rapid that its execution came at only a small cost in output state fidelity on the superconducting device. Such dynamic circuits are essential to future quantum-computing architectures as they will be needed, for example, to perform magic-state distillation circuits [7-9] and gate teleportation [36,37], and as well as many other measurement-based methods that have been proposed to complete a universal set of logic gates [11, 15-17, 38-49].”

“More work needs to be shown that backs the statement “each [single-qubit Pauli error] will be detected by some combination of the error sensitive events”

– One could use more discussion of how the adaptive operation would be compatible with errors that happen before it and after the stabilizer is measured to be -1.”

– For two-qubit gates, single faults can occur on two qubits at the same time, and a justification as to how the gadgets can tolerate them is in order alongside the single-qubit discussion in, e.g., Appx. E.”

The verification that we can detect the error events introduced by individual circuit elements was conducted by exhaustively simulating all possible insertions of local Pauli errors. This process is now explained in more detail in the corresponding Methods section, and the reader now is directed to this explanation in the main text.

We thank the author for the suggestion to also include two qubit Pauli errors for two qubit gates. This is now done using the same method as for single qubit gates, by exhaustive simulation. Again, the process is found to be fully fault-tolerant in these cases due to the flag circuits.

We add that the exhaustive search for errors was conducted using the variation of the error-suppression circuit with feedforward. This includes errors both before and after the feedforward operation. As such, our numerics verify the feedforward operation is robust to individual error events.

– In particular, the work could use more detail around the novel part of the magic-state circuit where one is required to be in the +1 eigenspace of S^Z .

We have added a discussion on the acceptance rates of the error-suppression cir-

cuits. This appears in the discussion of yield in the main text, and we elaborate on the details of this point in methods that we have added on yield. Notably, the new section explains how the feedforward operation increases the yield from a rate of $3/8$ up to $3/4$ in the noise free case, depending on access to a feedforward operation when we make the initial S^Z measurement.

“There are a few typos”

Many thanks for taking the time and care to identify these. We have corrected them all.

In response to referee #2:

We are grateful for your positive comments and constructive criticism. We respond to your comments as follows:

The manuscript is overall clearly written and can be followed well. However, the explanation of the context of the experiment should be strengthened. This is needed to make the relevance clearer to non-expert readers, who might not be experts in the variants of magic-state preparation and gate injection.

Many thanks for this comment. There are many details that need to be explained to put our experiment in the broader context of magic state distillation via state injection. To this end we have prepared supplementary information that broadly explains magic state distillation for the non-expert reader. Furthermore, we give details for how the encoded magic state we prepare can be injected into larger codes. We cover several example codes including the heavy-hex code. With the inclusion of this additional material, we have amended the abstract and introductory paragraphs to give a higher level picture of how our experiment might be used in large-scale quantum-computing architectures.

2) “In particular, how can one understand the quite strongly varying acceptance probabilities in the various logical bases? (see Fig. 4, from 15-30% to about 5%)?”

Let us respond to this comment together with the following related comment. We give our response below.

3) “Along a similar line: based e.g. on numerical simulations, can the authors comment on what the main bottleneck of the proposed protocol is? For example, it would be interesting to understand what the expected improvements over the observed improvement of about 4 would be as gate error rates and / or qubit measurement rates are changed. What are the expected performance gains from improving on these parameters in future, improved hardware?”

Thank you for these questions. It is indeed valuable to analyse the performance of the component we have demonstrated. We can estimate the strength of the noise by looking at the number of errors that are detected by our error checks. We can read the number of errors we detect from the yield plot; Fig. 4. We can study this using both numerical simulations and analytical modelling.

First, we find it instructive to construct a simple back-of-the-envelope calculations based on circuit noise model that explains the yield of different error-detection experiments. We present our model in a new Methods section.

In answer to question 2, our analytical model correctly explains the difference in acceptance rates for measuring different logical bases. There are two types of logical measurements, those using the measurement circuit in Fig. 2(b) and those using the circuit in Fig. 2(c). The latter is a longer circuit, and as such, there is more time for an error to be introduced to the system. This is captured by our model, shown in Eqn. (5). Indeed, Table IV shows tomography experiments conducted with circuit 2(b) have consistently better yields than those conducted with circuit 2(c) for the respective methods of preparation.

In regards to question 3, it is very difficult to characterise all of the sources of noise that occur throughout our experiment. Central to our analytical model is an error rate per parity measurement. We can comment on the successes and limitations of our model to get an indication on ways we can improve the performance of our component. Our model agrees reasonably well with experimental data if we assume an error rate per parity measurement $\sim 22\%$. As we explain in the new text, this is consistent with the experimental data if we conservatively assume that each of these circuit elements has an error rate of $\sim 2\%$. However, this is a very high error rate per gate compared with those we have estimated in Tables I and II. Our model on the other hand does not account for other noise processes such as leakage, cross talk, two-level systems as well as dephasing errors that occur during long idling periods while feedforward operations and measurements take place. As we discuss in the new text, it is very difficult to account for all of these processes by modelling or simulations.

Together with our analytical model we also present numerical simulations of our experiment assuming a circuit error model. As with the analytical model we find good agreement with the experimentally observed yield if we assume a gate error rate of around 2%. This lends further support to the success of the analytical model but, once again, this is a very conservative estimate for the gate error rate. Like our analytics, our numerical simulations do not account for other appreciable noise processes that are difficult to model in practice.

“A brief outlook of the work, possibly in the form of an additional methods / background section, would be important, again to highlight the context and what doors the present work opens.”

“Specifically, what are next or potential future steps that this specific demonstrated protocol opens? For example, can it be scaled up and how? Can this approach be generalised beyond the demonstrated fault-tolerant encoding in an error detecting code to error-correcting, possibly larger-distance codes? What kind of architectural constraints does this pose for the required quantum hardware? E.g. could the generation of Toffoli-states (and subsequent distillation of the latter) from these encoded CZ-states be achieved in a larger heavy-hexagon quantum processor architecture or what architectural adjustments or capabili-

ties would be generally required?”

We believe that the best way to use our scheme in a large-scale fault-tolerant quantum computer is to improve the fidelity of state injection for magic-state distillation protocols. We have elaborated on this in a supplementary document that we have included in our revised submission. It explains magic-state distillation at a higher level, showing how the CZ state can be converted into a Tofolli state, as well as how to inject the small code we prepare into larger codes for additional rounds of magic-state distillation. Among the codes we consider is the heavy-hex code. We give very specific details for how magic states can be injected onto larger codes using the device on which we conducted the experiment. As such, we present a clear path for how to use the component we have demonstrated in large-scale architectures that are now under development.

In answer to your questions regarding the architectures that will be needed to generalise the scheme to higher-dimensional codes, with our current understanding of quantum error-correction theory, we would need more complicated architectures that can realise three-dimensional codes, or more complicated two-dimensional architectures to perform the non-Clifford operations to prepare magic states on higher distance codes.

Other than developing hardware to realise these proposals that are already in the literature, it is not clear to any of the authors how to generalise our scheme for preparing magic states at present. Nevertheless, it could be interesting to look for ways of generalising magic state preparation with small and intermediate sized codes. We suggest a way of doing this is using pieceable fault tolerance. It may also be interesting to improve certain near-term methods of performing algorithms with error correction and error mitigation together. We have amended our discussion section with appropriate citations to mention this. It reads:

“We have shown that experimental progress has reached a point where we can make prototype gadgets that can impact the resource cost of large-scale quantum computers. In the accompanying Supplementary Information, we explain how our prototype can be used together with magic-state distillation. It will be exciting to continue to design, develop and test new gadgets with real hardware, that will improve the performance of the key subroutines needed for fault-tolerant quantum computing. Perhaps further developments in the theory of pieceable fault tolerance [45] might show us ways of producing better magic states with small devices. Error-suppressed magic states could improve the time cost of recent proposals [50,51] for error-corrected circuits that are supplemented by error mitigation techniques to complete non-Clifford operations. Ultimately, experimental progress we make to this end in the near term can benefit large-scale quantum-computing architectures.”

In response to referee #3:

We thank you for your positive recommendation and your constructive comments. We respond to your comments as follows:

What are the expected leakage rates on the devices and gates used in the experiment? Is there any quantitative data about leakage rates? Assuming leakage rates around 0.1% per gate or smaller, with the depth of the circuits used, accumulated leakage should not be a problem in a single round. However, is there enough wait time between subsequent rounds of experiment to ensure small enough probability of leakage to be present at the start?

Thank you for this question. Two-qubit gate-induced leakage rate has a median value of 0.033% on IBM Peekskill while the measurement-induced leakage rate for three ancilla qubits used in the main-text after one round of measurements has a median value of 0.49%.

We looked at the raw data taken with $100\mu\text{s}$ delay times in between, reclassifying I/Q for each qubit measurement and each round run on a handful of representative circuits (of varying depth) to detect leakage. From this analysis, we see on average 5.6% leakage present near the start of the each round, and 9.7% leakage at the completion of the circuit. These rates are consistent with the ballpark estimates from modelling leakage accumulation between rounds of circuit execution by taking into account the number of CNOT gates, mid-circuit measurements and resets, initialization and final readout, as well as the lifetime of the 2nd excited state.

There exists a trade-off between performing robust state tomography requiring fast execution of hundreds of circuits, and minimizing leakage rates by enabling a longer delay between circuit executions. For the results in the main text, the delay between subsequent circuit executions is $100\mu\text{s}$ for static circuits (no feedforward) and $50\mu\text{s}$ for dynamic circuits with hardware feedforward.

In all cases, error-suppressed and standard magic state preparations are run with identical settings. However, the error-suppressed magic-state circuit has a greater number of gates and measurements compared to the standard protocol. Based on circuit structure alone, the error-suppressed magic-state circuits may incur $\approx 1 - 3\%$ higher leakage accumulation than the standard case.

Do I understand correctly that in the post-selection version of the experiment, instead of the classical line on Fig. 2(a), the experiment is just stopped after an incorrect measurement result? Is the difference between the FF and PS state fidelities in Fig. 3 consistent with the increased experiment time, or are there additional error channels associated with the feedforward?

In the post selection (PS) case, the first S^Z measurement is performed and the classical measurement result is stored in memory. Subsequently, the W -measurement, and stabilizer checks are (always) performed. During post processing analysis, an additional selection-rule is applied which is that the first S_Z measurement must yield a +1 outcome for the PS case, else the result is discarded. In neither case is an experiment stopped in real-time after an incorrect measurement result.

The referee is correct to observe that the difference between state fidelities associated with FF and PS reflects not just an increase in execution time, but also different error channels in the following ways. Firstly, and as mentioned above, communication with the real-time control adds 500 – 700ns of idling time. It has been confirmed through independent experiments that the FF pathway is dominated by phase errors which, at present, cannot be suppressed through dynamical decoupling. Addressing this limitation in our control systems is a critical task for future work. Secondly, as discussed, the delay between repeated execution for the FF path is 50% shorter than for circuits without feedforward, leading to a difference in the susceptibility to leakage accumulation between rounds. Finally, classical readout errors on the control-wire may lead to the application of an incorrect recovery operation in the FF pathway. The noise channel associated with FF is not yet well understood. Indeed tools and benchmarks for appropriately characterizing non-unitary channels associated with measurement and feedforward are actively under development at IBM.

In response to this comment we have amended our comment describing the difference between the feedforward experiment, and the experiment completed with post selection as follows:

“The feedforward operations in our experiment can introduce idling periods, of the order of hundreds of nanoseconds, during which time additional errors can accumulate. To leading order we attribute the difference in fidelity between these preparation schemes to errors that occur while the control system is occupied performing the dynamical feedforward operation.”

What is the function of the H gate on line 4 in Fig. 5(b)? In general, I find the discussion about “removing all of the Pauli-X and controlled-not operations” a bit unclear; perhaps it would be better to give the original version of the circuit with these operations included?

Many thanks for this question. We see that we could have improved our explanation of this preparation circuit. The preparation circuit in Fig. 5(b) really serves two purposes; the first is to prepare the CZ state, and the second is to map the physical Pauli observables of the two qubits of the CZ onto logical Pauli observables of the error detecting code. This means that, when we measure the stabilizers of the code as in Fig. 5(a), the CZ state will remain in the code space of the target code. The Hadamard gate is necessary for this purpose. We have improved our description of this circuit as follows:

“In Fig. 5(a) we show a circuit that prepares an encoded CZ-state by, first, preparing a CZ-state on two physical qubits and, then, encoding the state such that the Pauli observables of the two qubits of the CZ state can be represented as logical operators of the error-detecting code we encode.”

For some additional clarity, we have also moved the Hadamard in the circuit so it lies with the ‘encoding steps’, rather than the CZ state preparation step.

In regards to the trivial elements of the circuit. These circuit elements are included in the figure. To make this discussion clearer we have added dashed lines to Fig. 5(b) to clearly delineate the trivial part of the circuit. It is also marked with an underbrace. We have also made reference to the trivial part of the circuit in the main text.

Minor corrections: page 3, left column, end of the 3rd line in the 1st complete paragraph, replace "rather they" with "rather than".

Many thanks for offering the time and care to spot this error. It has been fixed.

In Fig 2 caption, I would replace unconventional "stabilizer data" with a more commonly used "syndrome" or "syndrome bits".

We have made this change.

I believe the term "stabilizers" (albeit its common use) to be a bit colloquial, and suggest to use "stabilizer generators" instead (page 1, last paragraph on the right).

We have replaced the terms 'stabilizer' with either 'stabilizer measurement' or 'stabilizer operator' depending on the context in sentences where the term is used.

Reviewer Reports on the First Revision:

Referees' comments:

Referee #1 (Remarks to the Author):

I thank the authors for clarifying the broader context of their protocol. I did not fully appreciate of the interoperability of the $[[4,2,2]]$ code with both surface and color code state injection.

I agree with the authors that adaptive circuits are vital to a functioning fault-tolerant quantum computer. What is less clear is whether this **particular** adaptive protocol is an experimental/technological milestone.

I suggest the authors include the wider context behind their statements (as they have now done with the distillation protocol), summarizing previous literature on experimental achievements in feedback/control of superconducting circuits **in the context of error correction**.

For example, feedback has been implemented in superconducting circuits before [e.g., 1204.2479, 1302.5621, 1508.01385, 1508.01388, 1602.04768, 1709.01030, 1902.06946, 2107.11398, 2211.09116]. How is this circuit realization a substantial development over these and other prior approaches? Is this the first realization of an adaptive Clifford-type error-correction circuit on a multi-qubit system? Are there techniques, proprietary or otherwise, that have allowed this circuit to be realized to a degree of precision not possible before?

PS Please do not forget to update the arXiv.

Referee #2 (Remarks to the Author):

In the revised version of the manuscript, the authors have added a new Methods section E, showing results for the analytical model and from numerical simulations, with results shown in Table III. These are based on a simple, though experimentally motivated noise model that describes the observed experimental performance of magic state preparation well.

Furthermore, the authors have prepared extensive and clearly written supplemental information, in which they outline how Toffoli states can be obtained from the CZ states, which are fault-tolerantly prepared in this work. More importantly, they now outline protocols how the prepared magic state could be fault-tolerantly injected into leading candidate QEC codes for fault-tolerant quantum computing, including surface and color codes, as well as the heavy-hex code. The latter code can be directly hosted on the device on which the experimental magic-state preparation has been demonstrated. This additional material provides, in addition to some changes in the main text, the broader context and relevance, which was missing in the previous version of the manuscript. It further outlines convincingly the route via which the present experiment can fuel further advances towards scalable fault-tolerant universal quantum computing.

In summary, the authors have satisfactorily addressed my concerns. The results and improved version of the manuscript warrant publication in Nature.

The authors should still correct a number of minor typos I spotted:

- In the abstract: by producing a special resources called magic states
- Page 5: 'gate operations' in 'where mid-circuit measurements condition gates operations in real time'
- Suppl. Mat., page 1: known methods for distilling Tofolli states,. [2]
- Suppl. Mat., page 2, before Eq. (1): Tofolli in 'The Tofolli state is defined as follows:'
- Suppl. Mat., caption Fig. 3: in red in the bottom right corner. \diamond bottom left corner
- Suppl. Mat. Page 4: 'measuring the the stabilizer generators '
- Suppl. Mat. Fig. 7 caption: 'This can be peformed...'

Referee #3 (Remarks to the Author):

In my opinion, the authors of the manuscript 09629 "Encoding a magic state with beyond break-even fidelity", by Gupta et al, have convincingly answered all questions and criticisms, both from me and from other Reviewers. I strongly recommend publication of the manuscript in the current form.

Author Rebuttals to First Revision:

In response to referee 1

We are pleased that the referee is satisfied with our explanation of the wider applicability of our experiment in large-scale quantum-computing protocols.

Furthermore, we welcome their suggestion to expand the significance of our demonstration of feedforward. Our quantum control system is focused on scaling to larger numbers of qubits while maintaining general programmability to enable the exploration of real-time feedforward techniques. Our current devices have the ability to program feedback control with hundreds of qubits, and can scale to potentially thousands of qubits in future devices with no further architectural changes. The architecture supports arbitrary control flow and classical code execution within the coherence time of our superconducting qubit devices, such that we can obtain beyond break-even results in our experiment. Our modular platform enables us to explore quantum error correction in future work. In response to the referee's comment we have included a brief discussion of previous work in this area in the methods section and added some corresponding references.

In response to referee 2

We thank the referee for their positive recommendation, and for taking the care to identify typos in our supplemental material. We have corrected these typos accordingly.

In response to referee 3

We are very grateful to read this positive recommendation. We thank the referee for their constructive comments throughout this review process.